# Quorum sensing modulates the formation of virulent *Legionella* persisters within infected cells

Nicolas Personnic [1]*, Bianca Striednig [1], Emmanuelle Lezan[2], Christian Manske [3], Amanda Welin[1], Alexander Schmidt [2] & Hubert Hilbi [1]

The facultative intracellular bacterium *Legionella pneumophila* replicates in environmental amoebae and in lung macrophages, and causes Legionnaires' disease. Here we show that *L. pneumophila* reversibly forms replicating and nonreplicating subpopulations of similar size within amoebae. The nonreplicating bacteria are viable and metabolically active, display increased antibiotic tolerance and a distinct proteome, and show high virulence as well as the capacity to form a degradation-resistant compartment. Upon infection of naïve or interferon-γ-activated macrophages, the nonreplicating subpopulation comprises ca. 10% or 50%, respectively, of the total intracellular bacteria; hence, the nonreplicating subpopulation is of similar size in amoebae and activated macrophages. The numbers of nonreplicating bacteria within amoebae are reduced in the absence of the autoinducer synthase LqsA or other components of the Lqs quorum-sensing system. Our results indicate that virulent, antibiotic-tolerant subpopulations of *L. pneumophila* are formed during infection of evolutionarily distant phagocytes, in a process controlled by the Lqs system.

[1] Institute for Medical Microbiology, University of Zürich, Gloriastrasse 30, 8006 Zürich, Switzerland. [2] Proteomics Core Facility, Biozentrum, University of Basel, Klingelbergstrasse 50/70, 4056 Basel, Switzerland. [3] Max von Pettenkofer Institute, Ludwig-Maximilians University Munich, Pettenkoferstrasse 9a, 80336 Munich, Germany. *email: npersonnic@imm.uzh.ch

Phenotypic heterogeneity within clonal bacterial populations is a wide-spread strategy to colonize and persist in fluctuating environments[1]. An aspect of phenotypic heterogeneity of major clinical importance is the stochastic or stress-induced production of nonreplicating "persisters", which survive antibiotic exposure[2–7], but have not been thoroughly studied physiologically. The emergence of antibiotic-tolerant persisters has been documented for several important bacterial pathogens including *Staphylococcus aureus*, *Mycobacterium tuberculosis*, *Escherichia coli*, *Salmonella enterica* and *Pseudomonas* spp.[8]. The evolutionary origin of bacterial persistence and the extent to which this phenomenon is implicated in the ecology and environmental niches of pathogens remains unknown.

*Legionella pneumophila* is a ubiquitous environmental bacterium, which as an opportunistic pathogen can cause a severe pneumonia termed Legionnaires' disease. *L. pneumophila* replicates in a diverse array of protozoan hosts that comprise multiple phyla as well as in mammalian lung macrophages[9–12]. *L. pneumophila* survives ingestion by phagocytic cells by establishing a replicative membrane-bound compartment termed the *L. pneumophila*-containing vacuole (LCV)[13,14]. *L. pneumophila* employs the Icm/Dot type IV secretion system (T4SS) to inject a plethora of effector proteins, which promote LCV formation and prevent the fusion of the pathogen compartment with bactericidal lysosomes[15–20]. LCVs extensively communicate with the endosomal, secretory and retrograde vesicle trafficking pathways of the host cell and actively engage in the phosphoinositide (PI) lipid conversion from phosphatidylinositol 3-phosphate (PtdIns(3)$P$) to PtdIns(4)$P$[13,21–25].

At a population level, *L. pneumophila* employs a bi-phasic lifestyle, comprising a replicative phase and a postexponential, "transmissive" phase during which the bacteria are virulent and motile[26,27]. The switch between the replicative and transmissive phase, as well as a number of other traits of *L. pneumophila*, is regulated by the *Legionella* quorum-sensing (Lqs) system[28,29]. Components of the Lqs system comprise the autoinducer synthase LqsA, which produces the α-hydroxyketone signaling molecule LAI-1 (*Legionella* autoinducer-1, 3-hydroxypentadecane-4-one)[30], the membrane-bound sensor histidine kinases LqsS[31] and LqsT[32] and the prototypic response regulator LqsR[33], which dimerizes upon phosphorylation[34]. The bi-phasic lifestyle of *L. pneumophila* and a potential role of the Lqs system for infection have not been studied at single cell level.

In this study, we investigate the phenotypic heterogeneity of *L. pneumophila* in evolutionarily distant professional phagocytes. Using single cell techniques, we identify intracellular *L. pneumophila* nonreplicating persisters and further characterize their physiology. We reveal that the nonreplicating persisters are highly infectious and modulate their host cells to form a protective LCV. The nonreplicating subpopulation is of similar size in amoebae and interferon-γ-activated macrophages, and is controlled by the Lqs system.

## Results

**Intracellular *L. pneumophila* shows growth rate heterogeneity**. To explore whether a clonal population of *L. pneumophila* shows phenotypic heterogeneity within host cells, we investigated growth rate heterogeneity of single bacteria in their natural host, the free-living ameba *Acanthamoeba castellanii*. As a readout for the growth rate, we adapted for *L. pneumophila* the Timer[bac] system, a stable fluorescent reporter that slowly matures from a green to a red fluorescent protein[2]. Timer production did not impair the bacterial growth in broth or *A. castellanii* (Supplementary Fig. 1a). In exponentially growing *L. pneumophila* constitutively producing Timer (*L. pneumophila*/Timer), green

fluorescent Timer dominates over red fluorescent Timer, which is diluted by cell division before maturation, and the individual bacteria show a high green/red fluorescence (color) ratio [500 nm (green)/600 nm (red)] (Fig. 1a, 4 h, 8 h, and 12 h). By contrast, during the lag and stationary growth phases, *L. pneumophila*/Timer bacteria accumulate both green and red fluorescent Timer, and the individual bacteria show a low green/red color ratio (Fig. 1a, 2 h, and 24 h). Fluorescence ratios were robust in relation to cell-to-cell variations in protein content and cell size as indicated by the peaks' narrowness (Fig. 1a). To experimentally link the Timer color ratios to the division rates, we immobilized *L. pneumophila*/Timer in AYE/0.5% agarose[35], tracked growth over time by confocal microscopy, and measured the Timer color ratio at a single cell level (Supplementary Fig. 1b, c). Thus, we could equate fluorescence ratios ($R$) and the division rates ($\mu$) as follows [$\mu = \frac{0.1 + \text{Log10}R}{2.6}$], independently of the growth temperatures used (25 or 37 °C).

To assess intracellular growth rate heterogeneity, we infected *A. castellanii* with *L. pneumophila*/Timer and monitored the intracellular growth (Timer color ratio 500 nm/600 nm) at a single cell level by confocal microscopy. At 5 h post infection (p.i.), all *L. pneumophila* cells appeared red/orange (low green/red color ratio) indicating the absence of replication (Fig. 1b, Supplementary Fig. 2a and Supplementary Movie 1). At 24 h p.i., individual intracellular *L. pneumophila* showed various color ratios ($R$) (Fig. 1b, Supplementary Fig. 2a, Supplementary Movie 1) reflecting different intracellular bacterial division rates (Supplementary Fig. 1b) that ranged from growth arrest (NG, $R \approx -0.3$, $\mu = 0.0\,\text{h}^{-1}$) to slow growth ($G_S$, $R \approx 0$, $\mu \approx 0.04\,\text{h}^{-1}$) and fast growth ($G_F$, $R \approx 0.3$, $\mu \approx 0.17\,\text{h}^{-1}$). Bacteria with diverse growth rates were found in different amoebae as well as within the same host cell.

Flow cytometry analysis of lysates of *A. castellanii* infected for 24 h revealed the presence of fluorescent *L. pneumophila*/Timer subpopulations with even broader color ratios (Fig. 1c), the signal of which was specific and well separated from autofluorescent host cell debris (Supplementary Fig. 2b). Imaging flow cytometry analysis performed with lysates of infected amoebae confirmed the presence of *L. pneumophila* subpopulations with distinct Timer color ratio differences (Fig. 1d). Using the correlation between Timer fluorescence ratios and bacterial division rates defined by confocal microscopy analysis (Supplementary Fig. 1b), we estimated that intracellular *L. pneumophila* division rates ranged from 0 h$^{-1}$ to a maximum of about 0.4 h$^{-1}$ (Supplementary Fig. 2c). The average growth rate for the growing subpopulation was $0.17 \pm 0.02\,\text{h}^{-1}$, which represents a mean intracellular generation time of about 4 h.

Using the flow cytometry approach to quantify in a nonbiased manner the intracellular *L. pneumophila* populations, the nondividing subpopulation was surprisingly prominent, as it accounted for ~50% of all intracellular bacteria (Fig. 1e). Similar percentages of nondividing *L. pneumophila*/Timer subpopulations were obtained 24 h p.i., regardless of whether the inoculum comprised AYE-grown stationary phase *L. pneumophila* or bacteria "naturally" released from infected *A. castellanii* at the end of an infection cycle (Supplementary Fig. 2d). This result indicates that virulent *L. pneumophila* robustly forms a prominent (ca. 50%) subpopulation of nondividing bacteria upon infection of amoebae independently of the conditions previously encountered (broth or amoebae).

The *L. pneumophila* Δ*icmT* mutant strain lacks a functional Icm/Dot T4SS required to survive and replicate intracellularly in host cells[36]. Accordingly, the few Δ*icmT* bacteria surviving in amebae all showed a low green/red color ratio, indicating growth arrest (Fig. 1b–e, Supplementary Fig. 2c). By contrast the

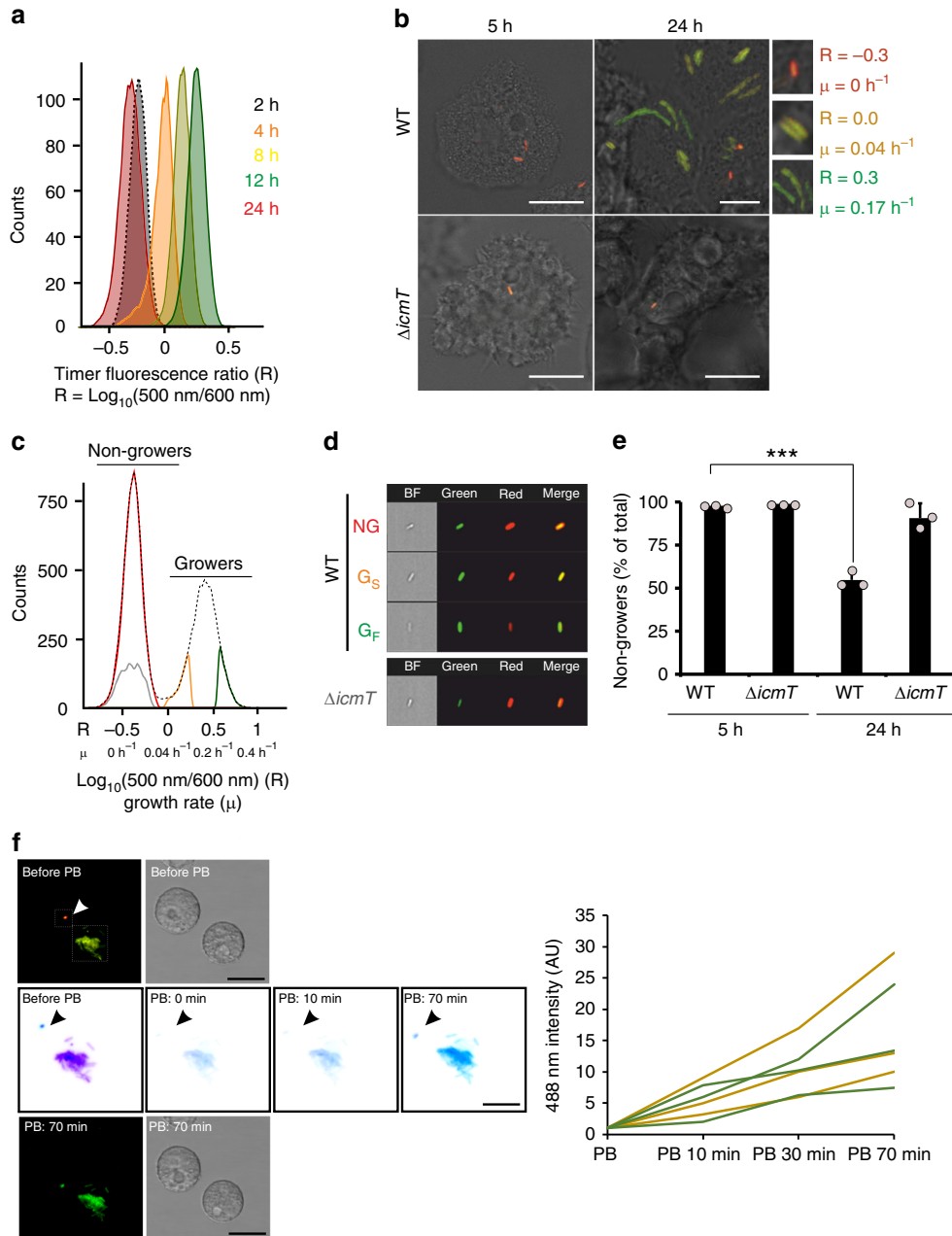

**Fig. 1** *L. pneumophila* shows growth rate heterogeneity in infected amoebae. **a** Timer color ratio reflects the division rate at a single cell level. Stationary phase grown *L. pneumophila*/Timer was diluted in AYE broth and let grow for 24 h. At given time points, bacteria were harvested and analyzed by flow cytometry. The Timer color ratio - Log$_{10}$[500 nm (green)/600 nm (red)] - was calculated for individual bacteria. **b**–**d** *L. pneumophila* intracellular growth rate heterogeneity. **b** Confocal microscopy of *A. castellanii* infected (MOI 1; 5, 24 h) with *L. pneumophila*/Timer wild-type (WT) or the isogenic avirulent Δ*icmT*/Timer strain. Micrographs show overlays of bright field and the Timer fluorescence (500 nm and 600 nm); growing and nongrowing bacteria appear green or red/orange, respectively. Magnifications: growth rate heterogeneity of *L. pneumophila* subpopulations (24 h p.i.) with different color ratios (R: Log$_{10}$[green/red] color ratio) and the corresponding division rate (μ). Scale bars 10 μm. **c** Flow cytometry or **d** imaging flow cytometry of lysed infected *A. castellanii* shows growth rate heterogeneity of released intracellular bacteria. Black, whole population; red, nongrowers (NG); orange, slow-growers (G$_S$); green, fast-growers (G$_F$). gray, Δ*icmT*. BF, Bright field. **e** *L. pneumophila* forms a high percentage of nongrowers in infected *Acanthamoebae*. Quantification by flow cytometry of nongrowing *L. pneumophila*/Timer wild-type (WT) or Δ*icmT*/Timer in infected cell lysates (MOI 1; 5, 24 h). **f** Intracellular nongrowers produce Timer protein de novo. *A. castellanii* was infected (MOI 1, 24 h), *L. pneumophila*/Timer and immobilized in PYG/0.1% agarose. Photobleaching (PB) was set up using the FRAP-wizard algorithm of the Leica SP8 microscope. Regions of interest (ROI; dashed squares) were photobleached (100% 488 nm laser intensity, iteration 30×), and fluorescence recovery at 500 nm was recorded (every 10 min). Micrographs illustrate FRAP kinetics in inverted color for clarity. Scale bar 20 μm. Fluorescence recovery measurements shown for three nongrowers (yellow) and three growing bacterial clusters (green). Data represent the mean ± SEM of three biological replicates (n = 3; light gray filled circles). Student's t test two-tailed, ***P < 0.001. Source data are provided as a Source Data file

complemented strain ΔicmT(icmTS) efficiently grew intracellularly and developed growing and nongrowing subpopulations (Supplementary Fig. 3a–c). Together, these findings demonstrate an extensive growth rate range of individual intracellular *L. pneumophila* in infected amebae, which requires a functional Icm/Dot T4SS. Moreover, our results also indicate that intracellular nongrowing bacteria comprise a substantial subpopulation.

**Intracellular nongrowers are viable and can resume growth.** Next, we addressed the question of whether the intracellular *L. pneumophila*/Timer nongrowers are viable. Propidium iodide-based live/dead staining performed on lysates of *A. castellanii* infected with GFP-producing *L. pneumophila* indicated that, regardless of the growth rate, 95% of the intracellular bacteria were viable at 24 h p.i. (Supplementary Fig. 4a, b). Moreover, confocal microscopy, flow cytometry, and imaging flow cytometry suggested, at the single cell level, that the intracellular nongrowers kept producing the green fluorescent Timer protein (Fig. 1b–d, Supplementary Figs. 1b, 2a–c, 3a, b, and Supplementary Movie 1). To further investigate the Timer protein production, we conducted fluorescence recovery after photobleaching (FRAP) experiments, which indeed confirmed that intracellular nongrowers synthesize Timer protein de novo (Fig. 1f). To assess another hallmark of bacterial viability, the membrane potential, we labeled intracellular *L. pneumophila* with the fluorescent mitochondrial membrane potential probe Mito-Tracker. 24 h p.i., MitoTracker equally stained growing and nongrowing intracellular *L. pneumophila*/Timer subpopulations, revealing the presence of a membrane potential (Supplementary Fig. 4c). Collectively, the ongoing translational activity, active metabolism, and an intact membrane potential indicate that intracellular nongrowing *L. pneumophila* are viable.

We further determined the growth capacity of intracellular *L. pneumophila* subpopulations by fluorescence activated cell sorting (FACS) and compared flow cytometry counts with colony forming units (CFU) on agar plates. To this end, we FACS-sorted *L. pneumophila*/Timer in lysates of *A. castellanii* infected for 24 h according to the Timer color ratios (Figs. 2a, b, 1c, Supplementary Fig. 2b, c). Intracellular growers and nongrowers were subsequently plated and CFU were determined. The plating efficiency (CFU per sorted fluorescent bacteria) was rather low, yet similar for the nongrowers and the growers (Fig. 2c), revealing that the growth capacity of the subpopulations was identical. Moreover, similar growth of the subpopulations in presence of chloramphenicol was observed, indicating that the plasmid expressing *timer* was not lost. To rule out the possibility that the observed growth resumption on plates was caused by cross contamination during the FACS-sorting, we assessed the growth resumption of sorted nongrowers by re-infection experiments and time lapse confocal microscopy. The re-infection approach revealed that sorted nongrowers efficiently (~85%) resumed growth in *A. castellanii* (Supplementary Fig. 5 and Supplementary Movie 2). We did not detect growth resumption of the sorted nongrowers upon immobilization in AYE/agarose, likely due to the low plating efficiency (Fig. 2c).

Finally, FACS-sorted growing and nongrowing isolates were first plated on agar plates and subsequently used to infect *A. castellanii* for 24 h. Under these conditions, the isolates passaged on agar plates developed similar percentages of growers and nongrowers in amoebae as determined by flow cytometry (Fig. 2d). These results reveal that the formation of growing and nongrowing subpopulations upon infection of amoebae is indeed a reversible phenomenon, and thus, a manifestation of phenotypic heterogeneity of clonal *L. pneumophila*. In summary, our findings show that the intracellular growth rate variation is a robust and reversible phenotype of viable *L. pneumophila*. Of note, the intracellular nongrowing subpopulation comprises viable and infectious individual bacteria.

**Intracellular nongrowers are nonreplicating persisters.** A hallmark of nongrowing bacterial subpopulations is their tolerance toward antibiotics[7,37,38]. We thus tested the sensitivity of the intracellular *L. pneumophila* subpopulations toward different classes of antibiotics. To this end, *A. castellanii* was infected with *L. pneumophila* for 24 h, the infected amoebae were lysed and FACS-sorted intracellular growers and nongrowers were exposed to the fluoroquinolone ofloxacin, the macrolide erythromycin or the aminoglycoside gentamicin (≥10× MIC[39–41]). The percentage of survivors was determined by comparing CFU to untreated subpopulations (Fig. 2e). Under these conditions, the nongrowing subpopulation showed a higher tolerance toward all tested antibiotics (ofloxacin > erythromycin > gentamycin). Alternatively, we exposed *A. castellanii* infected with *L. pneumophila* for 24 h p. i. to high concentrations of ofloxacin (300 μg mL$^{-1}$, 1 h) prior to FACS-sorting and CFU counting of the subpopulations. This approach confirmed that nongrowers survived antibiotics treatment better than growing bacteria also in vivo (Supplementary Fig. 6a, b).

The phenomenon of persistence is defined by a bi-phasic killing kinetic upon antibiotic exposure, where the majority of a clonal population is rapidly killed, while a subpopulation persists for a longer period of time[42]. To assess whether *L. pneumophila* forms persisters in host cells, *A. castellanii* was infected with wild-type bacteria for 24 h, the infected amoebae were lysed and the released bacteria were exposed to ofloxacin or erythromycin at a concentration of 100× MIC. Under these conditions, bi-phasic killing kinetics were observed for *L. pneumophila*, and the survivors exhibited similar kill curves in follow-up experiments (Fig. 2f), indicating the phenotypic rather than the genetic nature of the antibiotic tolerance. At the end of the antibiotic treatment, flow cytometry analysis performed in parallel to the plating confirmed that the survivors originated from a homogeneous population of nondividing bacteria (Supplementary Fig. 7a).

A possible mechanism for increased persistence upon treatment with antibiotics might be increased antibiotic efflux. We addressed the bacterial drug efflux capacity by measuring by flow cytometry the accumulation of ethidium bromide[43,44]. In agreement with a potential contribution to antibiotic tolerance, we measured significantly lower drug acquisition in FACS-sorted nongrowing compared with growing *L. pneumophila* in lysates of infected *A. castellanii* (Supplementary Fig. 7b). In summary, due to the bi-phasic killing kinetic of intracellular *L. pneumophila* upon exposure to antibiotic and the increased tolerance of the nongrowing subpopulation toward different antibiotics, intracellular *L. pneumophila* nongrowers are defined as nonreplicating persisters with increased drug efflux capacity (or reduced drug accumulation).

**The intracellular *L. pneumophila* subpopulations produce distinct proteomes.** To characterize the intracellular *L. pneumophila* subpopulations on a biochemical level, we analyzed nongrowers and growers by comparative proteomics. To this end, *A. castellanii* was infected with *L. pneumophila*/Timer for 24 h, lysed, and the FACS-sorted subpopulations were subjected to proteomics analysis. We identified more than 1000 proteins differentially produced in the growers and nongrowers (Fig. 3a, b and Supplementary Data 1). Among the growing subpopulation, slow- and fast-growers shared a very similar proteomic signature (Supplementary Fig. 8a–c, and Supplementary Data 2), and

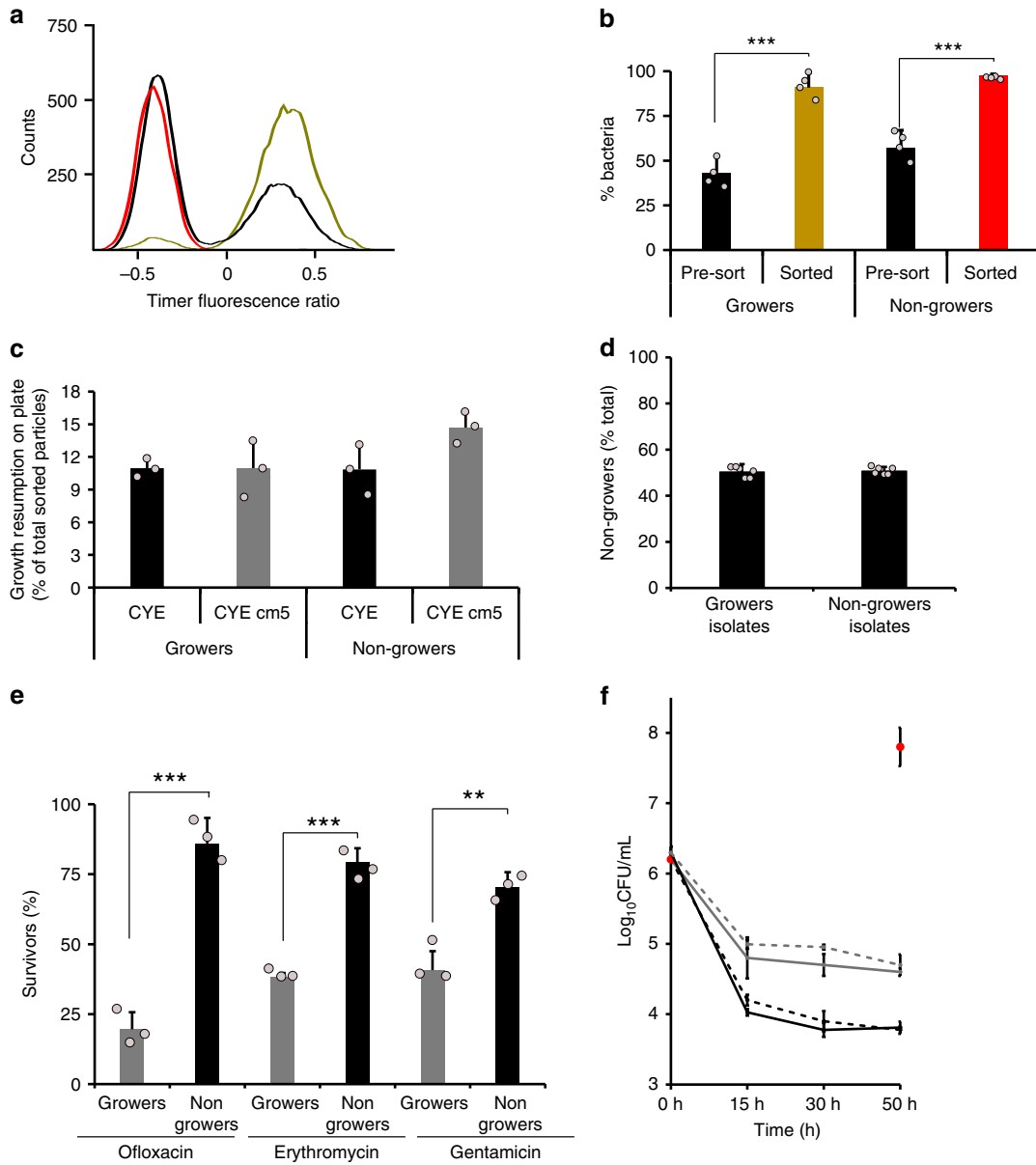

**Fig. 2** *L. pneumophila* intracellular nongrowers are viable and persisters. **a**, **b** FACS-sorting of intracellular *L. pneumophila* subpopulations. *A. castellanii* were infected with *L. pneumophila*/Timer (MOI 1, 24 h) and lysed. Released intracellular bacteria were FACS-sorted according to their Timer green/red color ratio, sorted subpopulations were reanalyzed by flow cytometry and compared with the initial infected cell lysate (pre-sort) to evaluate the separation efficiency. **a** Graph, (representative for four biological replicates) and **b** the quantification is shown. Pre-sort, black; growers, yellow; nongrowers, red. **c** Intracellular growers and nongrowers resume growth on CYE plates. To determine re-growth, FACS-sorted intracellular growing and nongrowing *L. pneumophila*/Timer subpopulations were plated on CYE or CYE/Cam plates and CFU were determined (plating efficiency: CFU*100/sorted fluorescent particles). **d** Growth rate heterogeneity is a reversible phenotypic trait. FACS-sorted intracellular growing and nongrowing *L. pneumophila*/Timer subpopulations were plated on CYE. Three colonies were harvested, grown to stationary phase in AYE broth and used to re-infect *A. castellanii* (MOI 1, 24 h). The size of the nongrowing population was determined by flow cytometry on infected cell lysates. **e** Ex vivo, nongrowers are more tolerant to antibiotic treatment. FACS-sorted growing and nongrowing *L. pneumophila* were treated (1 h) or not with the antibiotics ofloxacin (3 µg mL$^{-1}$), erythromycin (3 µg mL$^{-1}$) or gentamycin (20 µg mL$^{-1}$), and the percentage of survivors was calculated as $CFU_{treated}/CFU_{nontreated} \times 100$. **f** Formation of *L. pneumophila* persisters in infected amoebae. After host cell lysis, bacteria were resuspended in AYE supplemented with erythromycin (60 µg mL$^{-1}$; gray), ofloxacin (30 µg mL$^{-1}$; black), or without antibiotic (red dots). Bacteria were incubated at 25 °C and plated at defined time points. The antibiotics caused bi-phasic kill curves, indicating the existence of persisters. Survivors of 50 h of antibiotic treatment were harvested from plates and showed similar kill curves in follow-up experiments (dashed-lines). Data represent the mean ± SEM of at least three biological replicates (n ≥ 3; light gray filled circles). Student's *t* test two-tailed. ***P < 0.001, **P < 0.01. Source data are provided as a Source Data file

therefore, they were treated as one subpopulation of growing bacteria.

In the growing subpopulation of intracellular *L. pneumophila*, ribosomal subunits, aminoacyl-tRNA synthases, cell division proteins (e.g. FtsZ, SMC (structural maintenance of chromosomes) ATPases[45]), DNA replication proteins (e.g. DnaA, PolI) and DNA mismatch repair enzymes (e.g. MutS[46]) were enriched. These findings are consistent with an increased replicative activity of this subpopulation. Intracellular *L. pneu-mophila* growers also produced more IolD, an enzyme involved in

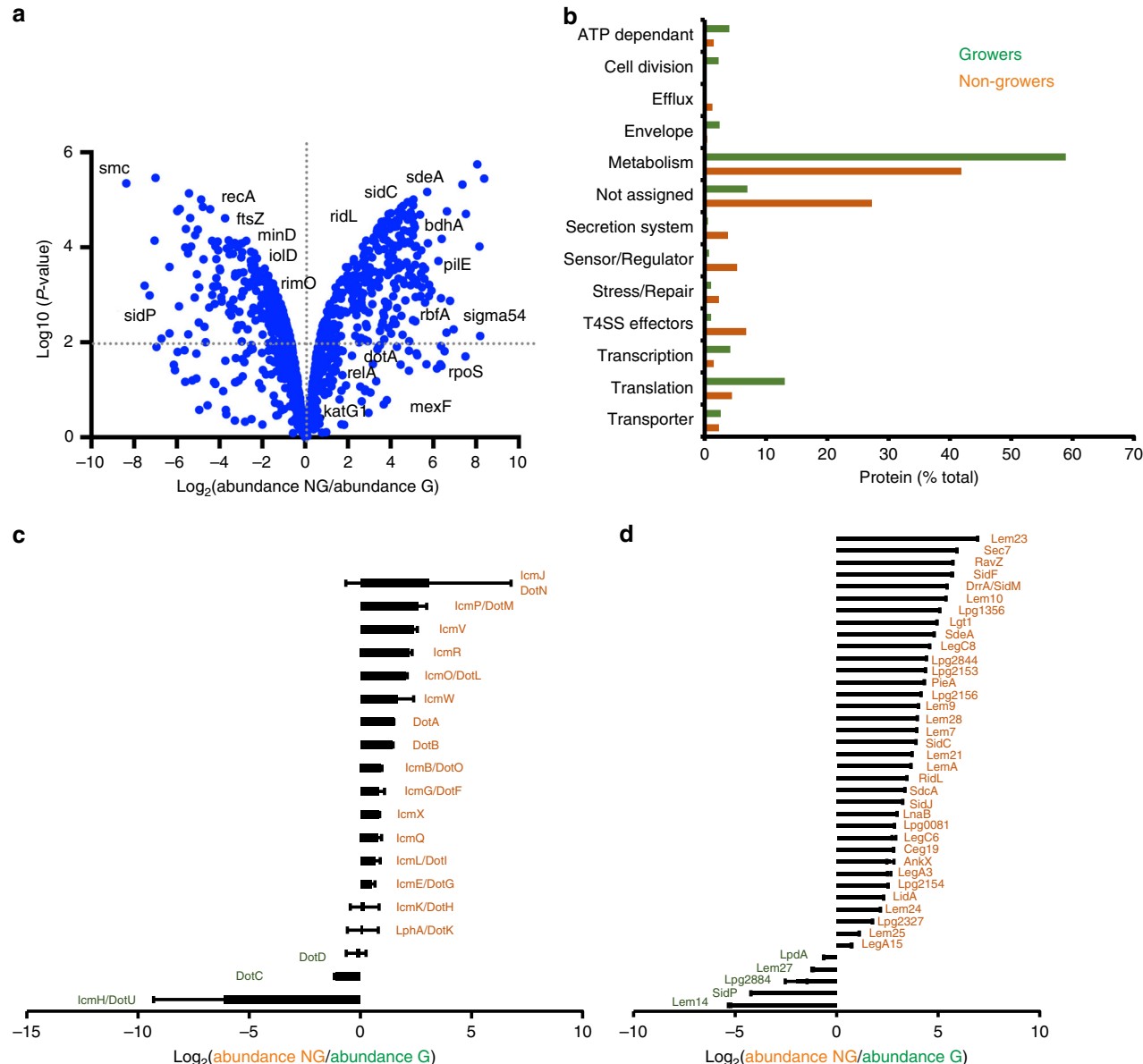

**Fig. 3** Intracellular *L. pneumophila* subpopulations produce distinct proteomes. **a** *A. castellanii* amoebae were infected (MOI 1, 24 h) with *L. pneumophila*/ Timer, lysed and the intracellular bacterial subpopulations FACS-sorted according to the Timer green/red color ratio. Comparative proteomics of sorted growers (G) and nongrowers (NG) revealed differentially produced proteins. Protein abundance in each subpopulation is depicted as volcano plot (see Supplementary Data 1 for the full dataset). **b** Functional classification of proteins identified in intracellular growing and nongrowing *L. pneumophila* subpopulations. The abundance of **c** Icm/Dot T4SS subunits and **d** Icm/Dot-translocated effector proteins are depicted. NG nongrowers, G growers. Data represent the mean ± SEM of four biological replicates ($n = 4$) pooled into two to increase the number of peptides detected

inositol catabolism, in agreement with previous findings on the intracellular catabolism of this carbohydrate by *L. pneumophila*[47]. Moreover, the intracellular growers preferentially produced the detoxifying enzyme Lpg2965, a peroxynitrite reductase, suggesting that they might undergo nitrosative stress.

The nongrowers preferentially produced the 3-hydroxybutyrate dehydrogenase BdhA and thus might feed on the storage compound poly-3-hydroxybutyrate. They also produced the lipase MhpC as well as the putative long chain fatty acid transporter Lpg1810, suggesting the catabolism of lipids. Interestingly, in broth, *L. pneumophila* utilizes exogenous fatty acid such as palmitic acid to synthesize poly-3-hydroxybutyrate[48]. Nongrowing *L. pneumophila* also produced the superoxide dismutase SodC, thereby responding to and being protected from reactive oxygen species. Along the same line, the

nongrowers further produced the DNA-glycosidase MutM that repairs ROS-mediated DNA damages[49]. Finally, as many as ~75% of the proteins of unknown function were preferentially produced by the intracellular nongrowing subpopulation (Fig. 3b). In summary, these findings suggest an alternative physiology and nutrient utilization for the growing and nongrowing intracellular *L. pneumophila* subpopulations.

**Intracellular *L. pneumophila* nongrowers are highly virulent**. In intracellular nongrowing *L. pneumophila* the sigma factor RpoN (σ$_{54}$) is primarily upregulated, but this subpopulation also produced the stationary phase sigma factor RpoS (σ$_{38}$,), which is implicated in virulence[50,51]. Moreover, the intracellular nongrowers preferentially produce the enzyme RelA involved in (p) ppGpp metabolism, indicating the activation of the stringent

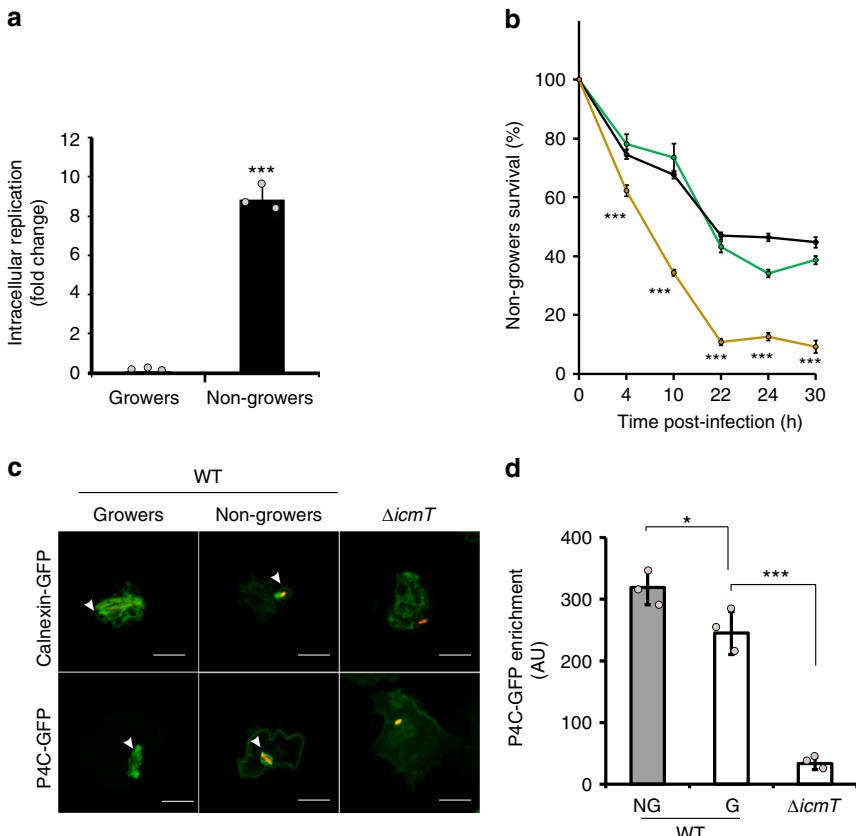

**Fig. 4** Intracellular *L. pneumophila* nongrowers are highly virulent. **a** Intracellular nongrowers are virulent. *A. castellanii* were infected (MOI 1, 24 h) with *L. pneumophila*/Timer, lysed and FACS-sorted according to the Timer green/red color ratio. Sorted growers and nongrowers were used to re-infect *A. castellanii*, and intracellular replication was determined by CFU (fold change: $CFU_{30\ min\ p.i}/CFU_{96\ h\ p.i.}$). **b** T4SS-dependent survival of intracellular nongrowers. *A. castellanii* were infected (MOI 1) with *L. pneumophila* wild-type (black), the isogenic avirulent Δ*icmT* strain (yellow) and the complemented strain Δ*icmT*(*icmTS*) (green) expressing *timer*. At given time points, the nongrowers from infected amoebae lysates were quantified by flow cytometry and analyzed by comparison with WT. **c**, **d** Intracellular nongrowers reside in individual LCVs. **c** *D. discoideum* amoebae producing the ER/LCV marker calnexin-GFP or the PtdIns(4)*P* probe P4C-GFP were infected (MOI 1, 24 h) with *L. pneumophila*/Timer WT or the Δ*icmT*/Timer strain, fixed and analyzed by confocal microscopy. Micrographs show overlays of the fluorescence at 500 nm (GFP, Timer) and 600 nm (Timer). White arrows show GFP-positive membrane only around intracellular WT growers and nongrowers. Scale bars 10 μm. **d** *D. discoideum*/P4C-GFP was infected (MOI 1, 24 h) with *L. pneumophila*/Timer WT or Δ*icmT*/Timer and P4C-GFP enrichment measured by imaging flow cytometry. Data represent the mean ± SEM of three biological replicates ($n = 3$; light gray filled circles). Student's *t* test two-tailed; ***$P < 0.001$, *$P < 0.05$. Source data are provided as a Source Data file

response in response to amino acid starvation, which is a hallmark of *L. pneumophila*'s switch from the replicative to the virulent/transmissive phase[52,53].

Strikingly, the comparative proteomics of intracellular *L. pneumophila* subpopulations revealed that the nongrowers upregulate components of the Icm/Dot T4SS (Fig. 3c, Supplementary Data 1) as well as 35 known Icm/Dot-translocated effector proteins (Fig. 3d, Supplementary Data 1). Most of those proteins are also produced when *L. pneumophila* is grown to the transmissive phase in broth[54]. To confirm the virulence of the intracellular nonreplicating persisters, we FACS-sorted the growing and nongrowing subpopulations that had formed in amebae 24 h p.i., subsequently reinfected *A. castellanii* and determined intracellular growth at 96 h p.i. by CFU (Fig. 4a). This approach revealed that the intracellular nongrowers were indeed highly virulent and also significantly more virulent than the growing subpopulation.

To further characterize the *L. pneumophila* intracellular nongrowers, we compared them to the Icm/Dot T4SS deficient Δ*icmT* mutant strain, which is defective for intracellular replication. To this end, lysates of amoebae infected for 24 h with *L. pneumophila* WT, Δ*icmT* or the complemented strain

Δ*icmT*(*icmTS*) producing Timer were analyzed by flow cytometry. This analysis revealed that a functional Icm/Dot T4SS significantly increased the survival of nongrowers over time (Fig. 4b). This result echoed with the incapability of *L. pneumophila* lacking a functional Icm/Dot T4SS to form a degradation-resistant compartment.

To assess, whether intracellular nongrowing *L. pneumophila* wild-type indeed localize in the unique calnexin- and PtdIns(4)*P*-positive LCV, we employed the genetically tractable ameba *Dictyostelium discoideum*, a versatile and powerful model of *L. pneumophila* infection[11]. In infected *D. discoideum*, *L. pneumophila*/Timer formed growing and nongrowing subpopulations of ca. 50% each 24 h p.i. (Supplementary Fig. 9), very similar to what was observed in infected *A. castellanii* (Fig. 1e). Fluorescence microscopy indicated that upon infection of *D. discoideum* producing the ER/LCV marker calnexin-GFP[55,56] or the Golgi/ LCV PtdIns(4)*P* probe P4C-GFP[57–59], with *L. pneumophila*/ Timer for 24 h, growing as well as nongrowing bacteria resided in membrane-bound compartments decorated with the GFP fusion proteins (Fig. 4c). Quantitative imaging flow cytometry analysis further indicated that growing as well as nongrowing *L. pneumophila* wild-type accumulated the PtdIns(4)*P* probe P4C-

GFP[59], in contrast to vacuoles harboring ΔicmT mutant bacteria, which are devoid of PtdIns(4)P (Fig. 4c, d). Altogether, these results indicate that intracellular nongrowers are highly infectious bacteria that require a functional Icm/Dot T4SS to survive in the host by forming distinct calnexin- and PtdIns(4)P-positive protective LCVs.

**Intravacuolar nongrowers upregulate genes of motility and virulence**. Upon entry into stationary growth phase in broth, *L. pneumophila* not only acquires virulence traits, but also becomes flagellated and motile[60]. As a proxy to assess the motility and virulence of intracellular nongrowing *L. pneumophila*, we monitored the expression of the gene encoding the major flagellum subunit, flagellin. To this end, we devised a fluorescent reporter combining the constitutive expression of *timer* and the transcriptional fusion of the gene encoding the blue fluorescent protein mCerulean with the promoter of *flagellin* (P$_{tac}$-timer−P$_{flaA}$-mCerulean, Supplementary Fig. 10a). As expected, the *L. pneumophila*(P$_{tac}$-timer−P$_{flaA}$-mCerulean) strain grown to stationary phase in AYE broth showed low Timer color ratios and produced mCerulean, in contrast to exponentially growing cultures (Supplementary Fig. 10b). We then infected *A. castellanii* with the *L. pneumophila*(P$_{tac}$-timer−P$_{flaA}$-mCerulean) strain and monitored the production of mCerulean at a single cell level by confocal microscopy (Fig. 5a). At 24 h p.i., intracellular subpopulations of nongrowing mCerulean-positive, and growing mCerulean-negative *L. pneumophila* had developed (Fig. 5a).

Next, we sought to validate the production of the Icm/Dot-translocated effector SidC by intracellular nonreplicating persisters (Fig. 3d). We thus exchanged the promoter of the *flagellin* gene with the promoter regulating the production of SidC (P$_{tac}$-timer−P$_{sidC}$-mCerulean, Supplementary Fig. 10a), and we confirmed the P$_{sidC}$ induction in stationary phase grown bacteria (Supplementary Fig. 10b). Upon infection of *A. castellanii* for 24 h with *L. pneumophila*(P$_{tac}$-timer−P$_{sidC}$-mCerulean), only intracellular nongrowers expressed *sidC*, as indicated by the production of mCerulean (Fig. 5b).

In order to quantify by flow cytometry the size of the intracellular *L. pneumophila* subpopulation that induces the *flaA* promoter, we constructed a dual fluorescent reporter comprising a constitutively expressed *mCherry* and a gene encoding an unstable GFP under the control of the promoter of *flagellin* (P$_{flaA}$-gfp, Supplemental Fig. 11a). In agreement with our previous findings, *L. pneumophila*/P$_{flaA}$-gfp grown in broth to stationary phase uniformly produced GFP (Supplemental Fig. 11b). Upon infection of *A. castellanii* for 24 h *L. pneumophila*/P$_{flaA}$-gfp developed a subpopulation of GFP-positive nongrowers (Fig. 5c). Flow cytometry analysis revealed that this GFP-positive subpopulation comprised ca. 45% of the total intracellular bacteria (Fig. 5d), which corresponds to the subpopulation size of intracellular nongrowers previously found (Fig. 1e). Similarly, upon infection of *A. castellanii* for 24 h with *L. pneumophila* harboring a transcriptional fusion of a gene producing an unstable GFP and the *sidC* promoter (P$_{sidC}$-gfp), GFP-positive nongrowers were observed 24 h p.i. (Fig. 5c). Flow cytometry further indicated that in infected amoebae lysates, the absolute numbers of GFP-positive bacteria harboring P$_{sidC}$-gfp or P$_{flaA}$-gfp were very similar (Fig. 5e).

Finally, infection of *D. discoideum* producing the Golgi/LCV PtdIns(4)P probe P4C-GFP[57–59] with the *L. pneumophila*(P$_{tac}$-timer−P$_{flaA/sidC}$-mCerulean) strains confirmed that the intracellular nongrowers producing mCerulean localize in degradation-resistant LCVs (Fig. 5f). In summary, these findings indicate that the intracellular nongrowers residing in replication-permissive LCVs upregulate hallmark genes of motility and virulence.

**L. pneumophila produces virulent persisters in macrophages**. Formation of intracellular nongrowers in infected amoebae incited us to evaluate the formation of nongrowers in macrophages, an evolutionarily distant phagocytic host for *L. pneumophila* and the primary target of the pathogen in the human lung[61]. Intracellular proliferation of *L. pneumophila*/Timer in macrophages required a functional Icm/Dot T4SS (Supplementary Fig. 12). Infection of mouse macrophages by *L. pneumophila*/Timer for 24 h revealed a nongrowing population that represented about 10% of the total intracellular bacteria (Fig. 6a, b), which is considerably smaller than the corresponding subpopulation in amoebae (Fig. 1e). Stimulation of macrophages with the pro-inflammatory cytokine interferon γ (IFN-γ) prior to infection by *L. pneumophila*/Timer for 24 h led to a fivefold increase in the number of intracellular nongrowers compared with infected naive macrophages, and reached ca. 50% of the total intracellular bacteria (Fig. 6a, b). Notably, this value was close to the nongrowing subpopulation size observed in infected *A. castellanii* (Fig. 1e). FACS-sorted nongrowers from lysates of infected IFN-γ-activated macrophages resumed growth upon infection of *A. castellanii* (Supplementary Fig. 13). Thus, nongrowing *L. pneumophila*/Timer released from activated macrophage are viable and infectious.

Next, we tested whether *L. pneumophila*/Timer forms persisters in naive or IFN-γ-treated macrophages. Upon exposure of lysates from infected naive macrophages to ofloxacin (100× MIC) bi-phasic killing kinetics were observed, indicating the presence of persisters (Supplementary Fig. 14). Interestingly, macrophage activation by IFN-γ triggered a significant increase in antibiotic-tolerant persisters (Supplementary Fig. 14), likely due to the higher portion of intracellular nongrowers as compared with naive phagocytes (Fig. 6a, b).

Using the *L. pneumophila*/P$_{tac}$-timer−P$_{flaA/sidC}$-mCerulean strains to infect macrophages, we further showed by confocal microscopy that the intracellular nongrowers were the only subset expressing hallmark genes of motility and virulence (Supplementary Fig. 15). For quantification by flow cytometry, we used the *L. pneumophila* P$_{flaA}$-gfp reporter strain, and determined that in naive macrophages about 10% of the total bacteria produce GFP under control of the P$_{flaA}$ promoter (Fig. 6c, d). INF-γ treated macrophages infected with *L. pneumophila* P$_{flaA}$-gfp (Fig. 6c, d) or P$_{sidC}$-gfp (Fig. 6e) reporter strains hosted five times more GFP producers as compared with the naive macrophages at 24 h p.i. In summary, these findings show that in naive macrophages and even more efficiently in INF-γ-activated macrophages, *L. pneumophila* forms infectious nonreplicating persisters expressing hallmark genes for virulence and motility.

**The Lqs system controls the formation of intracellular nongrowers**. Having characterized some features of the subpopulation of intravacuolar virulent persisters, we sought to identify genetic determinants controlling the intracellular phenotypic variation of *L. pneumophila*. We previously demonstrated that the Lqs system controls the switch from the replicative to the transmissive phase of *L. pneumophila*[28,29,33]. Moreover, a regulatory node bridging the Lqs system to the stringent response has been recently documented[62]. We thus hypothesized that the Lqs system may control the formation of intracellular virulent nonreplicating persisters.

Deletion of the LAI-1 autoinducer synthase *lqsA* led to a significant decrease (approximately twofold) in the formation of nongrowers in *A. castellanii* 24 h p.i., as compared with the parental strain (Fig. 7a, b). The Δ*lqsA*/Timer mutant was not impaired for virulence (Supplementary Fig. 16a) as previously reported for other Δ*lqsA* strains[31]. Thus, the reduction in

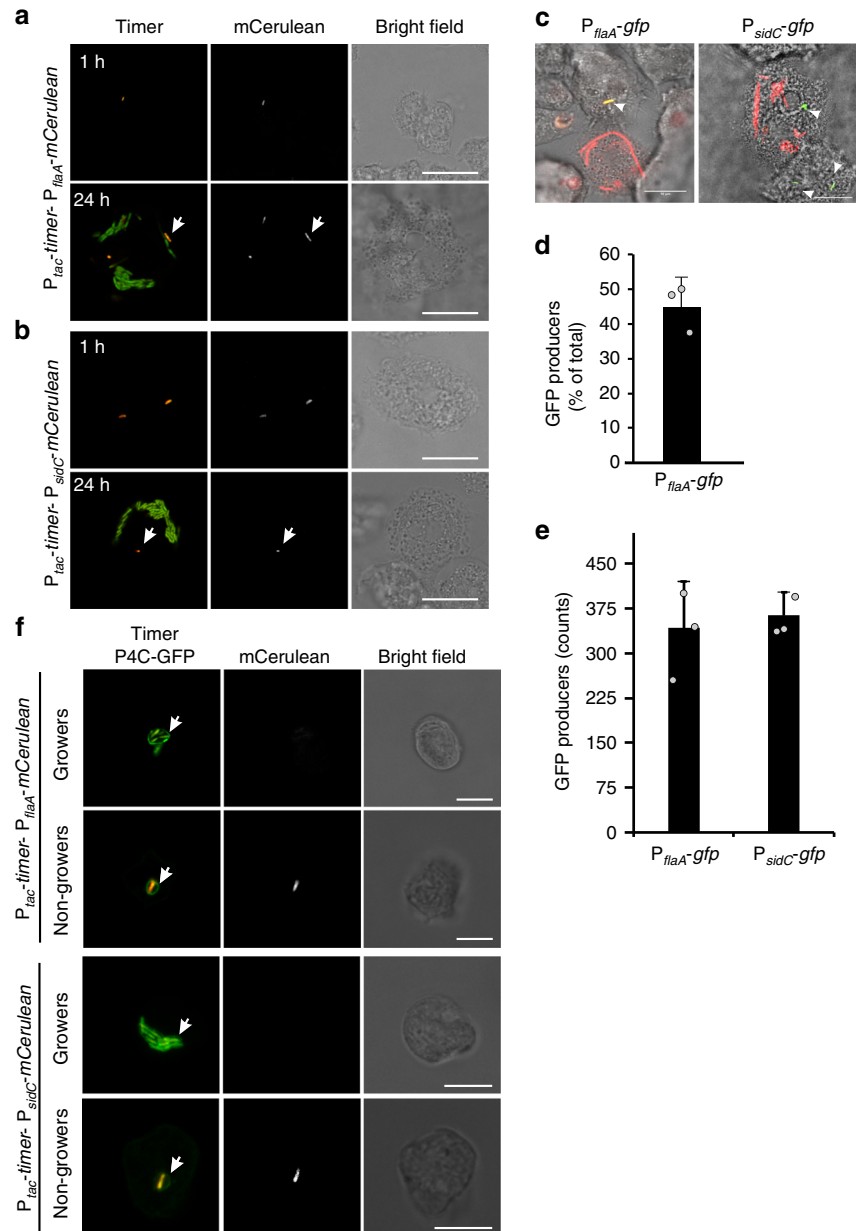

**Fig. 5** Intravacuolar nongrowers express hallmark genes of motility and virulence. *A. castellanii* was infected (MOI 1, 5 h or 24 h) with *L. pneumophila* harboring the fluorescent reporters **a** $P_{tac}$-timer−$P_{flaA}$-mCerulean or **b** $P_{tac}$-timer−$P_{sidC}$-mCerulean, fixed and analyzed by confocal microscopy. Micrographs show the fluorescence for Timer at 500 and 600 nm, for mCerulean at 479 nm and the bright field. The white arrows indicate intracellular red/orange nonreplicating and mCerulean-producing, transmissive bacteria. Scale bar 20 μm. **c** *A. castellanii* was infected (MOI 1, 24 h) with *L. pneumophila* harboring the fluorescent reporters $P_{flaA}$-gfp or $P_{sidC}$-gfp, fixed and analyzed by confocal microscopy. Micrographs show the overlay of bright field and fluorescence for $P_{flaA}$-gfp (500 nm, GFP; 600 nm, mCherry, all bacteria) and for $P_{sidC}$-gfp (500 nm, GFP; 405 nm, DAPI (false red color), all bacteria). The white arrows indicate intracellular nonreplicating and GFP-producing bacteria. Scale bar 10 μm. **d** The subpopulation fraction and **e** the absolute number of intracellular GFP producers in the infected cell lysates were determined by flow cytometry. **f** *D. discoideum* amoebae producing the LCV marker and PtdIns(4)*P* probe P4C-GFP were infected (MOI 1, 24 h) with *L. pneumophila* ($P_{tac}$-timer−$P_{flaA/sidC}$-mCerulean), fixed and analyzed by confocal microscopy. Micrographs show overlays of the fluorescence at both 500 and 600 nm (to detect Timer and P4C-GFP) and 479 nm (mCerulean). White arrows show GFP-positive membrane surrounding both intracellular growers and nongrowers. Only the nongrowers express hallmark genes for motility and virulence. Scale bars 20 μm. Data represent the mean ± SEM of three biological replicates (*n* = 3; light gray filled circles). Source data are provided as a Source Data file

nongrowers was not caused by an overall decrease of bacterial virulence, in contrast to what was observed for the avirulent Δ*icmT* mutant strain (Fig. 4b). This phenotype of the Δ*lqsA* mutant strain was genetically complemented (Fig. 7a, b) and also partially reverted upon co-infection with LAI-1-producing wild-type *L. pneumophila* (Fig. 7c). Finally, the Δ*lqsA* mutant strain formed reduced numbers of small colonies on CYE agar plates,

correlating with the reduced number of nongrowing cells in lysates from infected amebae (Supplemental Fig. 16b).

The sensitivity of the intracellular Δ*lqsA* mutant to antibiotics was tested with a >10× MIC of ofloxacin. This experiment revealed a lower antibiotic tolerance of Δ*lqsA* as compared with the parental strain (Fig. 7d). Furthermore, intracellular Δ*lqsA* harboring the fluorescent reporter $P_{flaA}$-gfp developed a reduced

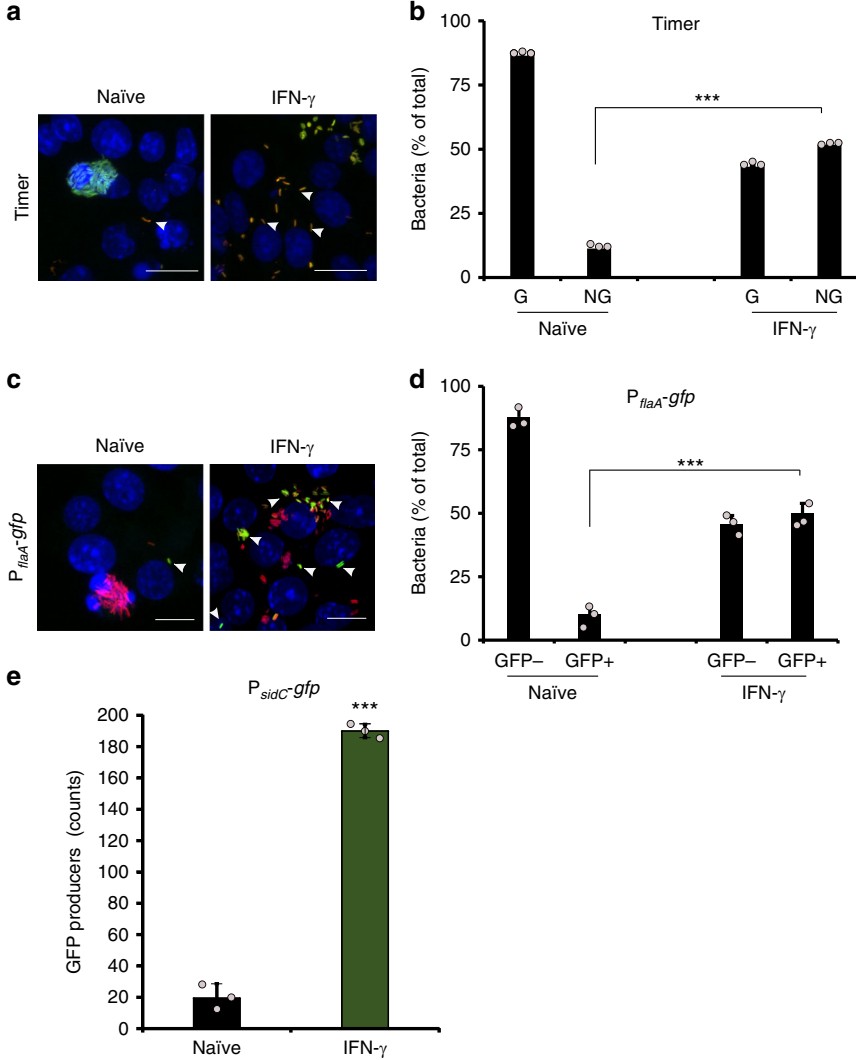

**Fig. 6** *L. pneumophila* produces intracellular nongrowers in macrophages. Naive and IFN-γ treated murine macrophages were infected (MOI 1, 24 h) with **a**, **b** *L. pneumophila*/Timer **c**, **d** *L. pneumophila*/P_{flaA}-*gfp* or **e** *L. pneumophila*/P_{sidC}-*gfp*, and **a**, **c** analyzed by confocal microscopy or **b**, **d**, **e** lysed and analyzed by flow cytometry. Micrographs show the overlay of Timer fluorescence (500 and 600 nm) and DAPI staining (nuclei). White arrows indicate intracellular nongrowers with low green/red Timer color ratio or producing GFP. Scale bars 10 μm. The histograms show the proportion of **b** intracellular growers (G) and nongrowers (NG) or **d**, **e** intracellular GFP producers (GFP+) and nonproducers (GFP-). Data represent the mean ± SEM of three biological replicates (*n* = 3; light gray filled circles). Student's *t* test two-tailed. ***P < 0.001. Source data are provided as a Source Data file

subpopulation of GFP producers (approximately twofold) in *A. castellanii* 24 h p.i. (Fig. 7e). Finally, the Δ*lqsA*(P_{sidC}-*gfp*) intracellular nongrowing subpopulation expressed the virulence gene *sidC*, as indicated by the production of GFP (Supplementary Fig. 16c). Together, these findings support the notion that LqsA controls intracellular growth heterogeneity and formation of *L. pneumophila* virulent nonreplicating persisters.

We also investigated whether the LAI-1 sensing and signal transduction pathway was contributing to the formation of nongrowers (Fig. 7f and Supplementary Fig. 17). Similar to the deletion of *lqsA*, the deletion of the individual LAI-1 sensor histidine kinases *lqsS* or *lqsT*[31,32] led to a reduction in nongrowers in *A. castellanii* at 24 h p.i. In contrast, deletion of both sensor kinases[32] even slightly increased the formation of intracellular nongrowers, and deletion of the response regulator *lqsR*[33] had no effect. The lack of phenotype for Δ*lqsR* prompted us to look for an alternative response regulator possibly controlling phenotypic heterogeneity. The transcription factor LvbR is negatively regulated by the sensor kinase LqsS[63]. Deletion of *lvbR* led to a loss in nongrowers in *A. castellanii* at 24 h p.i. (Fig. 7f and

Supplementary Fig. 17) The phenotypes of the mutants were reverted by genetic complementation (Fig. 7f). Of note, none of the tested mutants differed in its growth rate as compared with growing WT during intracellular growth (Supplementary Fig. 18a) as well as during growth in AYE broth (Supplementary Fig. 18b). In summary, these results indicate that the Lqs system controls the formation of intracellular *L. pneumophila* virulent nonreplicating persisters.

## Discussion

Using single cell approaches (fluorescence microscopy, (imaging) flow cytometry), we show here that the opportunistic pathogen *L. pneumophila* forms intracellular nongrowers in evolutionarily distant phagocytes (amoebae, macrophages). The intracellular nongrowing bacteria were viable and culturable, adopted increased tolerance toward antibiotics, and showed high virulence, as well as the capacity to form a degradation-resistant compartment in an Icm/Dot T4SS-dependent manner. Moreover, we found that the Lqs system controls the formation of the intracellular nonreplicating virulent *L. pneumophila* persisters.

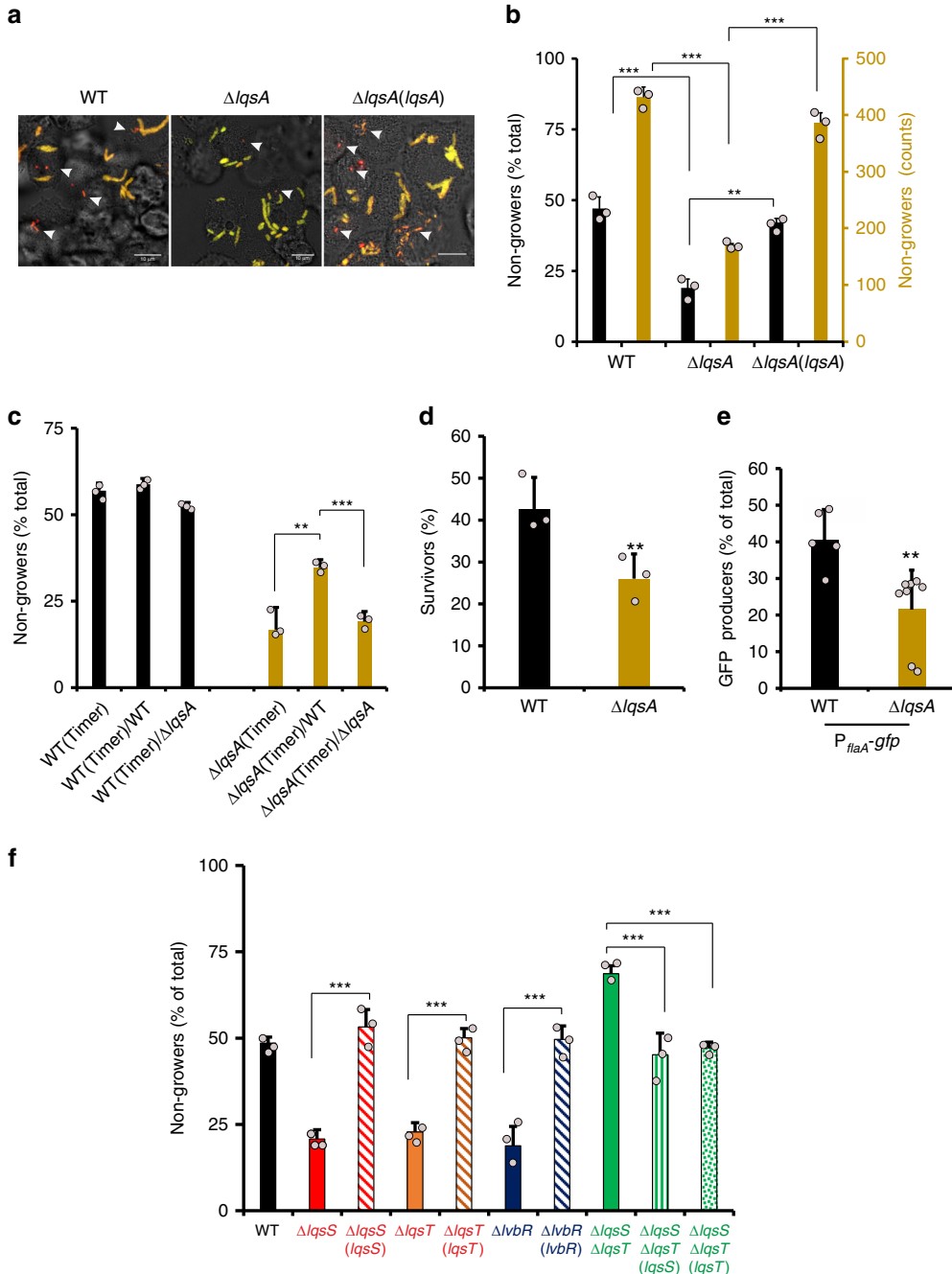

**Fig. 7** The Lqs quorum-sensing system controls the formation of intracellular nongrowers. **a–c** *lqsA* promotes the formation of intracellular nongrowers. *A. castellanii* were infected (MOI 1, 24 h) with *L. pneumophila* WT, Δ*lqsA* or the complemented strain Δ*lqsA*(*lqsA*) expressing *timer*, and (**a**) fixed and analyzed by confocal microscopy or (**b**) lysed and analyzed by flow cytometry. Micrographs show the overlay of bright field and Timer fluorescence (500 and 600 nm). White arrows indicate intracellular nongrowers. Scale bar 10 μm. Histograms in black show the subpopulation fraction of intracellular nongrowers (black, left *Y*-axis) and absolute numbers of nongrowers (NG) in cell lysates are shown (yellow, right *Y*-axis). **c** *A. castellanii* were co-infected (MOI 1, 24 h) with *L. pneumophila* WT and Δ*lqsA* (ratio 10:1), expressing *timer* or not, lysed and the proportion of intracellular nongrowers in lysates was determined by flow cytometry. **d** Deletion of *lqsA* increases the antibiotic sensitivity of intracellular *L. pneumophila*. *A. castellanii* were infected (MOI 1, 24 h) with *L. pneumophila* WT or Δ*lqsA* expressing *timer*, lysed and released bacteria were treated (1 h) or not with of ofloxacin (3 μg mL$^{-1}$). The percentage of surviving bacteria was determined by CFU. **e** Deletion of *lqsA* reduces the proportion in intracellular *L. pneumophila* expressing P$_{flaA}$. *A. castellanii* were infected (MOI 1, 24 h) with *L. pneumophila* WT or Δ*lqsA* expressing P$_{flaA}$-*gfp*, lysed and GFP producers were quantified by flow cytometry in lysates of infected amoebae. **f** The Lqs quorum-sensing system controls the formation of intracellular nongrowers. *A. castellanii* was infected (MOI 1, 24 h) with *L. pneumophila* WT, the isogenic Δ*lqsS*, Δ*lqsT*, Δ*lqsS*-Δ*lqsT*, or Δ*lvbR* mutant strains and the complemented strains Δ*lqsS*(*lqsS*), Δ*lqsT*(*lqsT*), or Δ*lvbR*(*lvbR*), Δ*lqsS*-Δ*lqsT*(*lqsS*), Δ*lqsS*-Δ*lqsT*(*lqsT*), expressing *timer*. After host cell lysis, the fraction of nongrowers was determined by flow cytometry. Data represent the mean ± SEM of at least three biological replicates (*n* ≥ 3; light gray filled circles). Student's *t* test two-tailed. \*\*\**P* < 0.001, \*\**P* < 0.01. Source data are provided as a Source Data file

Microorganisms enter nonreplicating persistence, either stochastically[4] or upon exposure to stress[2,5,6,64]. Research on intracellular nonreplicating persisters has recently gained heightened attention in efforts to tackle antibiotic tolerance and relapsing infections[7]. Using the natural interaction of *L. pneumophila* with protozoan predators, we reveal that the formation of intracellular nongrowing persisters is a pathogen strategy, which is applied to evolutionarily distant host cells, and which predates the emergence of metazoans. Intra-amebae nonreplicating persistence is a prominent phenomenon during amebae infection and may at least partly explain the difficulties to eradicate *L. pneumophila* in contaminated water systems[65].

Using comparative proteomics of FACS-sorted intracellular *L. pneumophila* subpopulations, we further show that the intracellular nonreplicating persisters remain metabolically active and might engage specific fatty acid-based metabolic routes. Proteins that have been shown to be important in persisters originating from other bacteria were elevated in the proteome of *L. pneumophila* intracellular nongrowers. Resistance-nodulation-cell division multidrug efflux pumps (e.g., MexF) and toluene resistance proteins were upregulated in the nongrowing subpopulation (12- and 5-fold, respectively). Intracellular nongrowers also produced fivefold more adenylate cyclase CyaA4 as compared with the growing subpopulation. An elevated level of the second messenger cyclic AMP (cAMP) correlates to increased antibiotic tolerance[66], which indeed was found for the intracellular nongrowing *L. pneumophila* population (Fig. 2e, Supplementary Fig. 6). In contrast, nongrowers produced lower amounts of the $F_oF_1$ ATP synthase (AtpH, AtpF, AtpD, AtpA2, and AtpG) in agreement with lower ATP concentrations present in nonreplicating bacteria and a link between bacterial nonreplicating persistence and decline in ATP production[5].

To survive in the bactericidal host, nonreplicating persisters subvert host functions using the Icm/Dot T4SS and build a protective, replication-permissive LCV. Recently, the subversion of cell autonomous immunity of infected macrophages by *Salmonella* nonreplicating persisters was reported[67]. Thus, manipulation of the host by intracellular persisters is likely to be a general and ancient trait. The proteomics results and their experimental validation indicated that nonreplicating persisters are fully equipped to initiate growth either in their current host cell or upon release. In agreement with this notion, we demonstrated that this intracellular subpopulation remains highly virulent for protozoan host cells (Fig. 4a, Supplementary Figs. 5, 13, and Supplementary Movie 2).

Virulence of *L. pneumophila* has been extensively studied at a population level[17,20,68] and may comprise different traits in exponentially growing and stationary phase bacteria in broth[54]. In this study, the comparative proteomics approach revealed the coexistence of distinct *L. pneumophila* intracellular subpopulations producing alternative sets of virulence factors, which thus might represent distinct aspects of intracellular *L. pneumophila* virulence. We notably identified a few virulence factors specifically enriched in the growing population (Fig. 3d, Supplementary Data 1), the functions of which in supporting intracellular growth remain unknown.

Infection of macrophages by *L. pneumophila* revealed a nongrowing population that represented about 10% of the total intracellular bacteria, which is considerably smaller than the corresponding subpopulation of about 50% in amoebae. A low number of intracellular nongrowers is in agreement with previous studies analyzing the facultative intracellular pathogen *S. enterica* serovar Typhimurium[2,6,69]. Intriguingly however, upon activation of the macrophages with IFN-γ, the nongrowing population increased to about 50% of the total intracellular bacteria. A possible interpretation of these observations is that the amoebae

behave as "activated" phagocytes for *L. pneumophila*. Macrophages activated by pro-inflammatory cytokine(s) show increased bactericidal activities. In addition, they promote the intracellular persistence and virulence of a bacterial pathogen.

In the current study, we identify the Lqs system as a genetic determinant controlling virulent *L. pneumophila* persister formation. The deletion of *lqsA* does not have a major impact on bacterial virulence[31], and only a few genes were regulated in *L. pneumophila* upon stimulation with the cognate signaling molecule, LAI-1[34]. Thus, major transcriptional alterations do not seem to underlie the observed phenotype. LAI-1-dependent signaling is complex and proceeds through the two homologous sensor kinases LqsS and LqsT, which are encoded by the *lqsS* gene located in the *lqs* cluster and the orphan *lqsT* gene, respectively[32]. The virulence and other phenotypes of the Δ*lqsS*-Δ*lqsT* double mutant strain are partially complemented by providing either *lqsS* or *lqsT* on a plasmid[32]. This feature is also seen for reversion of the phenotype of the Δ*lqsS*-Δ*lqsT* mutant regarding the increased ratio of growers vs. nongrowers (Fig. 7f). Moreover, the *lqsS* and *lqsT* genes are differentially regulated in stationary growth phase, and transcriptome studies indicated that 90% of the genes, which are downregulated in absence of *lqsT*, are upregulated in absence of *lqsS*[32]. The reciprocal and complex gene regulation pattern implicating signaling through LqsS and/or LqsT might also account for the fact that upon infection of *A. castellanii* with either the Δ*lqsS* or the Δ*lqsT* mutant strain the percentage of nongrowers was lower compared with wild-type *L. pneumophila*, while upon infection with the Δ*lqsS*-Δ*lqsT* double mutant strain the percentage was higher (Fig. 7f). Interestingly, the size of the nongrowing intracellular *L. pneumophila* subpopulation seems also to be controlled by the recently characterized DNA-binding transcription factor LvbR[63], rather than the response regulator LqsR[33]. LvbR links the Lqs and c-di-GMP regulatory networks[63], and the comparative proteomics approach documented in this study revealed that the intracellular nongrowing subpopulation shows a 10-30-fold enrichment in various diguanylate cyclases presumably producing c-di-GMP. Further studies will address the role of the second messenger c-di-GMP in regulating intracellular virulent *L. pneumophila* persisters. The work outlined here provides a basis to further analyze the triggers and consequences of intracellular phenotypic heterogeneity of a major human pathogen. In addition to its clinical relevance as the causative agent of a life-threatening pneumonia, *L. pneumophila* will serve as a paradigm to unravel on a molecular and cellular level the mechanisms of phenotypic heterogeneity of bacterial pathogens residing in phagocytic host cells.

## Methods

**Bacterial strains, eukaryotic cell lines, and reagents**. Bacterial strains and eukaryotic cell lines used in this study are listed in the Supplementary Data 3. *L. pneumophila* strains are derived from *L. pneumophila* JR32. They were grown for 3 days on charcoal yeast extract (CYE) agar plates, buffered with *N*-(2-acetamido)-2-aminoethane sulfonic acid (ACES) at 37 °C. Liquid cultures in ACES yeast extract (AYE) medium were inoculated at an $OD_{600}$ of 0.1 and grown fully aerated at 37 °C to stationary phase. Chloramphenicol (Cam; 5 μg mL$^{-1}$) was added as required. *A. castellanii* (ATCC 30234) was grown in proteose, yeast extract, glucose (PYG) medium at 25 °C or, when stated, in the fresh water mimicking Combo medium (UTEX media). *Dictyostelium discoideum* wild-type strain Ax3 producing calnexin-GFP[56,70] or P4C-GFP[59] was grown in HL-5 medium supplemented with geticin (G418, 20 μg mL$^{-1}$) at 23 °C. Nontumorigenic monocyte-derived Maf-DKO murine macrophages[2,71] were cultured in DMEM/10% FCS supplemented with 30% L-929 cell-conditioned medium in 5% $CO_2$. Cells were infected with *L. pneumophila* at a multiplicity of infection (MOI) of 1. After centrifugation at 170 × *g* for 10 min to enhance and synchronize the infection, cells were incubated for 45 min, subsequently washed and the culture medium replaced. For macrophages, 16 μg mL$^{-1}$ gentamicin for was added for 60 min to kill extracellular bacteria. Cells were subsequently washed and warm culture medium containing 16 μg mL$^{-1}$ added. When required, macrophages were pre-stimulated 24 h with 10 ng mL$^{-1}$ murine IFN-γ (PeproTech). Infected amoebae and murine macrophages were incubated at 25 °C and 37 °C respectively. MitoTracker Deep Red (Invitrogen) was

prepared and used according to the manufacturers' instructions. The use of MitoTracker as a bacterial membrane potential probe is described in ref. [72].

**Fluorescence reporters and complementation plasmids**. Plasmids and primers used are listed in the Supplementary Data 4 and 5. To construct the Timer growth rate reporter[2], the *timer* sequence was amplified from pBR322-timer (Addgene #103056) using the primers oNP46 forward/reverse, and subsequently was cloned into the pMMB207-C derivative pSN7, generating the plasmid pNP107. *timer* expression was under the control of the $P_{tac}$ promoter and the T7 ribosome binding site (RBS). Growth rate at a single cell level was defined by the Timer color ratio [(green, excitation 488 nm, emission 515–545 nm)/(red, excitation 561 nm, emission 600–620 nm)]. Nongrowing bacteria have a low Timer color ratio and appear red/orange. Growing bacteria have a high color ratio and appear green. To generate $P_{flaA}$-*gfp* (dual fluorescence reporter) the locus harboring both *mCherry* and the short-live *gfpsfm2-laa* from pTSARUd2.4s[73] was cloned into the pMMB207-C derivative pCM4[34] using the primers oNP26 forward/reverse to generate the plasmid pNP83. Constitutive *mCherry* expression was achieved by inserting the $P_{tac}$ promoter from pNT28[33] using the primers oNP41 forward/reverse, yielding the plasmid pSN1. *mCherry* expression was improved using the T7 RBS amplified from pNT28 using the primers oNP45 forward/reverse, generating the plasmid pSN2. To increase GFPsfm2-LAA degradation rate[74], the sequence encoding the peptide tag AANDENYAAAV at the carboxyl terminus was generated by site-directed mutagenesis of pSN2 using the primers oNP28 forward/reverse (Quik-Change II site-directed mutagenesis; Agilent) yielding the plasmid pSN5. A hairpin transcriptional terminator followed by a multiple cloning site was synthesized and inserted upstream the gene encoding *gfpsfm2-aav*, yielding the plasmid pSN6. It allows a quick promoter exchange and prevents polar transcriptional activity originating from the $P_{tac}$ promoter. Promoters of interest were mapped by combining published promoter maps, RNA sequencing[75] and the software Softberry (http://www.softberry.com/) available online. For $P_{flaA}$-*gfp*, the *flaA* promoter was amplified from the *L. pneumophila* chromosome using the primers oSN2 forward/reverse and inserted upstream the *gfpsfm2-aav* to generate the plasmid pSN7. Plasmid-based expression systems are a source of fluctuations caused by differences in plasmid copy numbers or by alternative protein production capacities within the bacterial population. mCherry was used to normalized GFP production in order to accurately quantify changes in the promoter activity as well as to detect bacteria by microscopy or flow cytometry. To generate $P_{sidC}$-*gfp* the *sidC* promoter was amplified from the *L. pneumophila* chromosome using the primers oCM78 forward/reverse and inserted into pCM4[34] upstream of the unstable *gfp* to generate the plasmid pCM11. To generate $P_{tac}$-*timer*–$P_{flaA/sidC}$-*mCerulean, mCerulean* was amplified from pNP99 using the primers oNP72 forward/reverse and used to replace the *gfp* in pCM9[34] and pCM11. The corresponding new plasmids are pNP125 and pNP126, respectively. The locus $P_{flaA/sidC}$-*mCerulean* was then amplified from pNP125 and pNP126 using the primers oNP73 forward/reverse and inserted into pNP107 at the BspQI site. The plasmids thus generated were named pNP127 and pNP128, respectively. To construct a complementation plasmid for the Δ*icmT* mutant strain, the locus $P_{tac}$-*timer* was amplified from pNP107 using the primers oNP77 forward/reverse and inserted into pGS-Lc-37-14[36] leading to plasmid pNP124. To construct a complementation plasmid for the Δ*lqsA* mutant strain, the *lqsA* gene was amplified from the *L. pneumophila* JR32 chromosome using the primers oNP60 forward/reverse and inserted into pNP107, downstream of the *timer* encoding gene, yielding the plasmid pNP120. *timer* and *lqsA* are organized as an operon. To construct a complementation plasmid for the Δ*lqsS*, Δ*lqsT*, and Δ*lvbR* mutant strain, The locus $P_{tac}$-*timer* was excised from pNP107 using the restriction enzyme ApaI and XmnI and inserted into pNT31[31], pAK2[32], and pAK18[63] leading to the plasmid pNP121, pNP122, and pNP123, respectively.

**Comparative proteomics**. For sample preparation ~$60 \times 10^6$ *A. castellanii* were infected for 24 h, yielding $15 \times 10^6$ sorted bacteria per subset and replicate (four replicates). Lysates and sorted bacteria were kept in ice-cold PBS supplemented with 180 μM Cam to prevent protein degradation and synthesis. Sorted bacteria were lysed in 50 μl of lysis buffer (1% sodium deoxycholate (SDC), 10 mM TCEP, 100 mM Tris, pH 8.5) using 20 cycles of sonication (one cycle is 30 s on, 30 s off; Bioruptor, Dianode). Then, proteins were reduced at 95 °C for 10 min and, after cooling to room temperature, the proteins were alkylated in 15 mM chloroacetamide for 30 min at 37 °C. Proteins were digested using sequencing-grade modified trypsin (1/50, w/w, trypsin/protein; Promega, USA) overnight at 37 °C. After digestion, the samples were supplemented with TFA to a final concentration of 1%. Peptides were cleaned up using PreOmics Cartridges (PreOmics, Martinsried, Germany) following the manufacturer's instructions. After drying the samples under vacuum, the peptides were resuspended in 0.1% aqueous formic acid solution at a concentration of 0.5 mg mL$^{-1}$. 0.5 μg of peptides of each sample were subjected to LC–MS analysis using a dual pressure LTQ-Orbitrap Elite mass spectrometer connected to an electrospray ion source (both Thermo Fisher Scientific) as recently specified[76] and a custom-made column heater set to 60 °C. Peptide separation was carried out using an EASY nLC-1000 system (Thermo Fisher Scientific) equipped with a RP-HPLC column (75 μm × 30 cm) packed in-house with C18 resin (ReproSil-Pur C18–AQ, 1.9 μm resin; Dr Maisch GmbH, Ammerbuch-Entringen, Germany) using a linear gradient from 95% solvent A (0.1% formic acid, 99.9% water) and 5% solvent B (80% acetonitrile, 0.1% formic

acid, 19.9% water) to 35% solvent B over 50 min to 50% solvent B over 10 min to 95% solvent B over 2 min and 95% solvent B over 18 min at a flow rate of 0.2 μl/min. The data acquisition mode was set to obtain one high resolution MS scan in the FT part of the mass spectrometer at a resolution of 240,000 full width at half maximum (at 400 m/z, MS1) followed by MS/MS (MS2) scans in the linear ion trap of the 20 most intense MS signals. The charged state screening modus was enabled to exclude unassigned and singly charged ions and the dynamic exclusion duration was set to 30 s. The ion accumulation time was set to 300 ms (MS1) and 25 ms (MS2). MS1 and MS2 scans were acquired at a target setting of $10^6$ ions and 10,000 ions, respectively. The collision energy was set to 35%, and one microscan was acquired for each spectrum. For label-free quantification, in our LFQ workflow the MS raw files were imported into the Progenesis QI (Nonlinear Dynamics, v2.0) and analyzed using the default parameter settings. MS2 data were exported directly from Progenesis in mgf format and analyzed using Mascot (Matrix Science, version 2.4.1), against a concatenated target-decoy database containing normal and reverse sequences of the predicted SwissProt entries of *L. pneumophila* (ATCC_33152/DSM_7513, www.uniprot.org, release date 9/05/2017), and commonly observed contaminants (in total 10,006 protein sequences) generated using the SequenceReverser tool from the MaxQuant software (Version 1.0.13.13). The Mascot search criteria were set as follows: 10 ppm precursor ion mass tolerance, 0.6 Da fragment ion mass tolerance, full tryptic specificity required (cleavage after lysine or arginine residues unless followed by proline), maximum of three missed cleavages, fixed modifications, carbamidomethylation (C); variable modification, oxidation (M) and acetyl (protein N-term). Results from the database search were imported into Progenesis and a list with all quantified peptides exported. The quantitative data were further processed and statically analyzed using the SafeQuant software tool[77]. In brief, the false-discovery rate (FDR) of identifications on the peptide and protein level was set to 1% based on the number of decoy hits obtained from reversed protein sequence entries. For quantification, the analysis included global data normalization by equalizing the total peak areas across all LC–MS runs, summation of peak areas per protein and LC–MS2 run, followed by calculation of protein abundance ratios. Only isoform specific peptide ion signals were considered for quantification. The summarized protein expression values were used for statistical testing of between condition differentially abundant proteins. Here, empirical Bayes moderated t-tests were applied, as implemented in the R/Bioconductor limma package.

The resulting per protein and condition comparison *p*-values were adjusted for multiple testing using the Benjamini–Hochberg method (http://bioconductor.org/packages/release/bioc/html/limma.html). The mass-spectrometry proteomics data have been deposited to the ProteomeXchange Consortium via the PRIDE[78] partner repository with the dataset identifier PXD015106 (project https://doi.org/10.6019/PXD015106).

**Confocal microscopy and image processing**. Sixty minutes after inoculation, infected amoebae were seeded into chambered coverslips (ibidi μ-Slide8 Well). At given time points, infected cells were washed 3× with PBS and fixed with 2% PFA/0.1% glutaraldehyde (Electron Microscopy Sciences) for 60 min. Following fixation the cells were washed and the fixative was quenched with 0.1 M glycine for 20 min at room temperature. Cells were immobilized by adding a layer of PBS/0.1% agarose. Alternatively, cells were analyzed by time lapse microscopy. Image acquision was performed using the confocal microscope Leica SP8 at 63× magnification. Image processing was realized with the ImageJ software. Timer color ratio at a single cell level was obtain as follow: [(green, excitation 488 nm, emission 515–545 nm)/(red, excitation 561 nm, emission 600–620 nm)].

**Correlation of Timer fluorescence ratio and growth rates**. *L. pneumophila* microcolony formation is described in reference[35]. Briefly, stationary phase grown bacteria were embedded in AYE/0.5% agarose at a final $OD_{600\,nm}$ of 0.1 and poured into chambered coverslips (ibidi μ-Slide 8 Well). After solidification, single bacteria were let grown into microcolonies for 16 h (at 25 °C) or 12 h (at 37 °C). Microcolony formation was monitored by confocal microscopy. The Timer color ratio was measured at a single cell level and correlated to the number of division that occurred.

**Fluorescence recovery after photobleaching**. For FRAP experiments, *L. pneumophila*-infected amoebae were seeded 60 min p.i. into chambered coverslips (ibidi μ-Slides 8 Well). Twenty-four hours p.i., the infected cells were immobilized by adding a layer of PYG/0.1% agarose. Fluorescence acquisition was performed using the confocal microscope Leica SP8 at 63× magnification. Photobleaching was set up according to the FRAP-wizard of the Leica user interface on narrowed regions of interest. Photobleaching was performed using the FRAP-booster, 100% 488 nm laser intensity for 79 ms and 30× iterations.

**Flow cytometry analysis**. Phagocytes were infected with *L. pneumophila* wild-type or the isogenic deletion mutant strains and lysed using 0.1% Triton TX-100 (Sigma) in 150 mM NaCl (amoebae) or PBS (macrophages). After centrifugation the pellets were washed in PBS and fixed with 2% PFA/0.1% glutaraldehyde for 60 min. Following fixation cells were washed and the fixative was quenched with 0.1 M glycine for 20 min at room temperature. Relevant spectral parameters were

subsequently recorded in a FACS-Fortessa II. The gating strategy was performed as described in references[2,7,3] for the Timer and the dual fluorescence reporters, respectively. Data processing was realized with the software FlowJo. Spectral properties collected using the Timer and dual fluorescence reporter were analyzed by calculating the $Log_{10}[(Ex\ 488\ nm, Em\ 515–545\ nm)/(Ex\ 561\ nm, Em\ 600–620\ nm)]$ for each detected bacterium. To accurately determine the fluorescent particle counts and to be able to compare conditions, we used defined resuspension volume and acquisition time.

**Intracellular bacterial viability**. *A. castellanii* infected for 24 h with *gfp* expressing *L. pneumophila* was lysed using 0.1% Triton TX-100 (Sigma) in 150 mM NaCl. Released bacteria were washed in PBS and incubated with propidium iodide (PI; 10 µg mL$^{-1}$, 30 min). Bacterial viability was subsequently determined by flow cytometry. A gating strategy using the GFP signal was used to identify the bacteria in the lysate. Bacterial death was triggered by fixing the lysate with 4% paraformaldehyde (PFA, Electron Microscopy Sciences) and served as PI positive controls. The use of PFA was guided by the need of preserving GFP fluorescence and bacterial shape.

**Imaging flow cytometry**. For imaging flow cytometry analysis of PtdIns(4)*P* localization to the LCV, *D. discoideum* Ax3 producing the PtdIns(4)*P* probe P4C$_{SidC}$-GFP were infected (MOI of 1) with *L. pneumophila* expressing *timer*. The cells were detached and fixed with 2% PFA (Electron Microscopy Sciences) for 60 min, and the fixative was quenched with 0.1 M glycine for 20 min at room temperature. At least 20,000 events were acquired using an ImageStream X MkII imaging flow cytometer (Amnis). Data analysis was performed with IDEAS 6.2 software, and after color compensation, at least 600 cells with internalized red fluorescent bacteria were identified by gating and classified as containing one or multiple bacteria as previously described[59]. In the gated cell populations, the mean intensity of P4C$_{SidC}$-GFP on one LCV or several LCVs was analyzed in a two pixel-wide mask surrounding the bacteria using the feature [Mean Pixel_Dilate(Spot (M04, Lpn, Bright, 8.5, 1), 4) And Not Dilate(Spot(M04, Lpn, Bright, 8.5, 1), 2) _P4C]."

**Cell sorting**. *A. castellanii* infected for 24 h with *timer* expressing *L. pneumophila* were lysed using 0.1% Triton TX-100 (Sigma) in HS buffer (20 mM N -2-hydroxyethylpiperazine-N-2-ethanesulfonic acid; 250 mM sucrose; 0.5 mM ethyleneglycoltetraacetic acid; pH adjusted to 7.2 with 1 M KOH). Lysates were centrifuged (250 × *g*, 15 min), and resuspended in PBS, followed by nine passages through a ball homogenizer (Isobiotech, http://www.isobiotec.com) using an exclusion size of 6 µm. Samples were sorted according to the Timer color ratio using an Aria II (BD Biosciences) with scatter and fluorescence channels (green, excitation 488 nm, emission 515–545 nm; red, excitation 561 nm, emission 600–620 nm), a nozzle size of 70 µm, using the four-way purity mode and a sorting efficiency >90%. Sorted subsets were systematically reanalyzed.

**Persister assays**. Bi-phasic kill curves: ~10$^7$ phagocytes were infected as described above. After 24 h, infected cells were lysed using 0.1% Triton TX-100 (Sigma) in HS buffer and the bacteria were resuspended in AYE supplemented or not with erythromycin 60 µg mL$^{-1}$ (>100× MIC) or ofloxacin 30 µg mL$^{-1}$ (>100× MIC). Bacterial suspension were incubated at 25 °C or 37 °C depending on the host cell used. At given time points, bacteria were collected and washed three times. For each wash 1 mL PBS was used and 950 mL of the supernatant discarded. The bacterial suspensions were subsequently serially diluted 1:10 and plated to quantify CFUs. Survivors were harvested from the plates from the last time point and used for follow-up experiments.

**Antibiotic tolerance of subpopulations ex vivo and in vivo**. Approximately 10$^7$ *A. castellanii* were infected as described above. After 24 h, amoebae were lysed and the released intracellular growing and nongrowing bacteria were sorted according to the Timer color ratio for each individual. Sorted subpopulations were subsequently incubated in fresh AYE supplemented or not with antibiotics (≥10× MIC; 1 h, 3–20 µg mL$^{-1}$ depending on the antibiotic) or ethidium bromide (1 h, 1 µg mL$^{-1}$). After washing, antibiotic tolerance for each subpopulation was evaluated by plating the bacteria and quantifying CFU. Ethidium bromide uptake was monitored by flow cytometry analysis. Alternatively, infected cells were treated or not with highly concentrated ofloxacin (300 µg mL$^{-1}$) for 1 h. Infected cells were subsequently lysed and 10$^5$ released intracellular growing and nongrowing bacteria were sorted according to the Timer color ratio for each condition. Antibiotic tolerance for each subset was evaluated by plating the bacteria and quantifying CFU.

**Growth resumption of intracellular nongrowers**. Approximately 10$^7$ *A. castellanii* or IFN-γ-treated murine macrophages were infected as described above. After 24 h, the phagocytes were lysed, the released intracellular nongrowing bacteria were sorted according to the Timer color ratio and used to infect fresh *A. castellanii*. Intracellular growth resumption of the sorted nongrowers was monitored by confocal microscopy.

**Quantification and statistical analysis**. Statistical differences were determined using a two-tailed Student *t* test on the means of at least three independent experiments. Probability values of less than 0.05, 0.01, and 0.001 were used to show statistically significant differences and are represented with *, **, or ***, respectively. For the comparative proteomics, the summarized protein expression values used for statistical testing of between condition differentially abundant proteins. Empirical Bayes moderated *t*-tests were applied, as implemented in the R/Bioconductor limma package

The resulting per protein and condition comparison *p*-values were adjusted for multiple testing using the Benjamini–Hochberg method (http://bioconductor.org/packages/release/bioc/html/limma.html).

**Reporting summary**. Further information on research design is available in the Nature Research Reporting Summary linked to this article.

## Data availability

The mass-spectrometry proteomics data have been deposited to the ProteomeXchange Consortium via the PRIDE partner repository with the dataset identifier PXD015106. All other relevant data are included in this article and its Supplementary Information files, or from the corresponding author upon request.

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

## Acknowledgements

We thank members of the Hilbi group for comments and discussions. We are grateful to Prof. Francois-Xavier Campbell-Valois for generously providing pTSARUd2.4 s and Selina Niggli for initial cloning. N.P. and B.S. research in the laboratory of H.H. was supported by the Swiss National Science Foundation (SNF) Ambizione program (PZ00P3_161492 & PZ00P3_161492 185529) awarded to N.P. Research in the laboratory of H.H. was supported by the SNF (31003A_153200, 31003A_175557), the Novartis Foundation for Medical-Biological Research, the OPO Foundation, and the German Research Foundation (DFG; SPP 1617). A.W. was supported by a grant from the Swedish Research Council (2014-396). The funders had no role in study design, data collection and analysis, decision to publish, or preparation of the paper.

## Author contributions

N.P. conceived the study with input from H.H. and designed the experiments. N.P., B.S., E.L., C.M., A.W. and A.S. performed the experiments. N.P. and H.H. wrote the paper with input from all authors.

## Competing interests

The authors declare no competing interests.
