## [Peer Review File · Nature Communications]

Reviewers' comments:

Reviewer #1 (Remarks to the Author):

"Quorum sensing elicits a subpopulation of vacuolar virulent *Legionella* persisters", by Personnic and colleagues describes an experimental investigation of growth heterogeneity in *Legionella pneumophila* populations that had infected amoeba and murine macrophages. Using the Timer protein, the authors observed two distinct subpopulations in terms of fluorescence, which they assessed were growing and non-growing cells. The intracellular non-growing cells remained culturable and they exhibited higher tolerance to 3x MIC concentrations of antibiotics over short treatment periods (1 hr). The authors then measured the proteomes of growers and non-growers and discussed several differences. The non-growers happened to be quite virulent upon amoeba re-infection experiments, which was quite interesting. The authors found that the abundance of cells that expressed fluorescent virulence markers approximated the size of the population that was non-growing inside amoeba, and they observed similar phenomena in murine macrophages. Genetically, the authors implicated the type 4 secretion system, *lqsA* (an autoinducer synthase), both autoinducer sensory histidine kinases (*lqsS* and *lqsT*), and transcription factor *LvbR*. Overall an important topic with some interesting single cell data, and my concerns are detailed below.

1. To claim that the non-growing cells are persisters, higher antibiotic concentrations need to be used and biphasic kill curves need to be presented. Persister measurements are routinely conducted at very high antibiotic concentrations (sometimes up to 100x MIC) to prevent measurements from reflecting the abundance of spontaneously-resistant mutants. Biphasic kill curves are also a requirement, which means survival needs to be measured as a function of time (not just at one time point) and two clear regimes must be observed when a log-linear plot of survival vs time is produced. The authors might find the following review helpful (PMID: 27080241). The authors should also assess whether survivors from those antibiotic treatments exhibit the same survival kinetics during a follow-up experiment (which has typically been done to show the phenotypic rather than genetic nature of persistence).

2. Line 44: Persisters are not resistant, they are tolerant to antibiotics.

3. Line 94: Did they authors mean "-0.3" for growth arrest. 0.3 appears to be within the growing population.

4. The y-axis in 2b should go down to 0%, starting at 30% is a bit misleading.

5. Fig 2C: how can the CFU per event be higher than 100%? Somewhat lower than 100% is reasonable since some events might be caused by noise for instance, but why would it be over 100%? More CFUs than events that were sorted?

6. Line 178, I believe the authors mean "nitrosative"

7. Line 179-189: Stating that those over-abundant proteins are markers of bacterial persistence is not accurate. Unless the cited research was done in *L. pneumophila*, which it was not, there is no evidence that efflux pumps, cyclic AMP, ATP, or other proteins discussed are markers of persistence in *L. pneumophila*.

8. Line 162-200: None of the proteins mentioned in this section about comparative proteomics were actually investigated further. As far as I can tell, all of the statements in this section are speculative and not confirmed or explored with additional assays. It seems that this section would be better suited for the discussion, since they are discussing potential reasons why they would observe that data and not going further. The section after this one with T4SS hits from the proteomics were followed up with additional assays, which justifies its placement in the results section.

9. In Figure 4c, I can't see the wild-type growing cells being in those fluorescently labeled vacuoles. Perhaps because they are also labeled with GFP? Could a different fluorescent protein (not green or red, which Timer uses) be used for the vacuoles?

10. When looking at the *sidC* and *flaA* GFP reporters, why didn't the authors change the fluorescent protein color so that they could assign cells as growing or non-growing with Timer, and then assess their expression of virulence factors in the same cells? Providing data where the abundance of cells that express those factors is close to the abundance of cells that are non-growing is not direct evidence of the connection between those two phenotypic outputs and growth status.

11. Figure 6f: can the authors provide an explanation as to why deletion of both sensory kinases produced the opposite effect of the single deletions?

12. Complementation of the *lqsA* deletion mutant was well received, but complementation of the other mutants with phenotypes would be fitting (*icmT*, *lqsS*, *lqsT*, *lvbR*).

Reviewer #2 (Remarks to the Author):

The manuscript by Personnic et al. "Quorum sensing elicits a subpopulation of vacuolar virulent *Legionella* persisters" describes how the non-replicating virulent persisters of *Legionella* in phagocytes are controlled by the Lqs system and what are the differences between the replicating and non-replication populations in the phagocytes and amoebae. I was asked to comment specifically on the proteomics part of the study.

It is difficult to be sure whether the study is technically sound because the data is shown in a way that precludes such insight. There are several modifications at least to the data presentation that have to be made before the manuscript can be reconsidered.

1. The raw data should be deposited in a publicly available database (such as ProteomeXchange) and the dataset access should be clearly described.

2. It was difficult to see if there were any replicates, even though one could assume there were because the statistical analysis was performed. The only information about the three replicates was in the legend for the Supplemental Figure 2. It should be clearly stated in the Methods section.

3. Were all the proteins listed in the tables identified and quantified in all three replicates? The tables should list the numbers for each replicate separately and then the averages and p-values.

4. Ultimately, the results of the proteomic analysis do not seem to show anything surprising or novel, but mainly confirm the known features of different subpopulations, or at least the data is presented that way.

Reviewer #3 (Remarks to the Author):

Legionellosis occurs worldwide and outbreaks are mainly caused by waterborne *Legionella pneumophila*, which is fatal if not treated promptly, and is the focus of this manuscript. *L. pneumophila* is ubiquitous in the environment, and can multiply both within aquatic protozoa and in macrophages. Moreover, the growth and viability of *L. pneumophila* can vary under environmental fluctuations, such as different temperatures.

The authors use a Timer fluorescent reporter system, constitutively expressed from an episomal

plasmid, to monitor the growth state of *L. pneumophila* during infection of amoebae and of naïve and activated murine macrophages. The authors infer that *L. pneumophila* derived either from amoebae or from activated macrophages produce about 50% of non-growing cells. The authors also tested whether the non-growing bacteria were viable, metabolically active, antibiotic susceptible, and virulent, and they made a comparative analysis of the proteome of both growing and non-growing subpopulations extracted from amoebae. They claim that phenotypic heterogeneity of growth among intracellular bacteria is reversible in vitro. They also conclude that non-growing bacteria tolerate better antibiotic treatment, express persistence-related proteins and are more virulent. At the end of the manuscript the authors, relying partly on their proteomic data and partly on the literature, create some *L. pneumophila* mutant strains, striving to understand the genetic origin of the intracellular formation of persistent variants, which remains open to speculation.

In conclusion, the topic addressed in this manuscript is scientifically relevant and ambitious. Nonetheless, the fluorescent reporter system used, on which almost all the experiments were based, was not sufficiently characterized. This makes it difficult to draw clear conclusions in a systematic manner, also because of the lack of some adequate controls. This raises major concerns, which are detailed below, that need to be addressed in order to support the main conclusions of the manuscript.

1. The Timerbac system was well characterized in *Salmonella* (Claudi et al., Cell 2014) and used as a growth-rate reporter. Although the Timerbac system is very attractive, several factors can influence its expression, including oxygen. Timerbac should be used cautiously in a new species, notably when the bacterium is versatile as to environmental changes, such as temperature. The authors should provide a careful characterization of this fluorescent reporter system in their model organism. In particular, the correlations between the growth rate and green and red fluorescence over the whole growth curve in vitro (OD and CFUs) and at different temperatures used to grow amoebae and macrophages (25° C and 37° C) are lacking. For instance, Fig. 1a shows the relationship between the Timer Log10 color ratio and two growth phases, without providing information regarding the actual division rate.

2. Since the G/R ratio was not clearly associated to specific growth rates, how were the non-growers exactly defined (Fig. 1d, 4, 5, 6, Supplementary Figures 1a, 3, etc.)? For instance, in lines 596-97 and 409 and in SuppInfo line 28, the authors claim that non-growing bacteria appear red, whereas the bacteria look mostly yellowish/orange as they still express green fluorescence. Could they be slow growers and what is the intracellular growth rate of cells that turned green? The definition of NG, Gs and Gf should be carefully determined in vitro first and then applied to the infection models.

3. Moreover, since Timerbac was expressed from an episomal plasmid, it is important to assess to what extent the variable number of copies of the plasmid contributes to the observed heterogeneity.

4. It is also unclear how the bacteria behaved during the first 5 hours of infection (Fig 1b and Movie) and whether they were already non-growing prior to infection, or if they transitioned to a non-growing state upon infection.

5. A reference for the Δ icmT strain is missing, as well as a comparative growth curve between *L. pneumophila* WT, Δ icmT and the strain overexpressing Timerbac.

6. To determine the viability of non-growing *L. pneumophila* the authors assessed the membrane potential by staining bacteria with a far-red dye, which is mainly used to label mitochondria. Indeed the BacLigh Bacterial Membrane Potential Kit would be a more specific tool, even though not compatible with Timerbac. Moreover they claim that the presence of green fluorescence in non-growing cells, which are often defined as red, is an indicator of active metabolism. Being the

Timerbac reporter expressed from a strong constitutive promoter, the green fluorescence, even in the absence of growth, could depend on the accumulation of older protein, which is not matured into its red form, due to the absence of metabolic activity. A FRAP experiment should help solving this issue. In addition, non-growing cells might be dead even though they continue to glow with fluorescence, because old protein expressed from a strong promoter can endure in the body of dead cells. In sum, it is crucial to perform counter-staining to spot live versus dead cells, and this must also be associated with the sorting experiments throughout the manuscript.

7. Although Fig. 2b shows a good sorting efficiency, the gating strategy with regard to both a non-fluorescent control strain and the sorted subsets must be specified, and the same number of events per subset should be shown in Fig. 2a and Supplementary Figure 2.

8. In the sorted subset of growers (Fig. 2a) there is a fraction of cells that has a fluorescence ratio comparable to the non-growers. How did the authors make sure that the growth resumption found in non-growers (Fig. 2c,d) was not due to a contamination from growers and vice versa?

9. In lines 137-140 the authors claim that the formation of growing and non-growing subpopulations is a reversible phenomenon, and that the sorted subsets developed similar percentages of growers and non-growers. Next, in lines 145, 146, 628 and 632, the authors state that they tested the sensitivity of "intracellular" *L. pneumophila* to antibiotics, and claim that "intracellular" non-growers are non-replicating drug persisters. However, the persistence assay was not carried out on "intracellular" bacteria but on subpopulations extracted from the host, which apparently recover a similar phenotype in vitro. To test which of the two subpopulations survives drug treatments best, the authors should directly treat the intracellular bacteria with antibiotics and extract and sort them only after treatment. Therefore lines 152, 153, 159, 160 and 179 are not corroborated by the experiments.

10. The authors propose that increased antibiotic efflux in non-growers is likely to cause antibiotic persistence. However, it seems more rational that cells that do not grow are less efficient at internalizing compounds, as shown in Fig. 2f, rather than at extruding them more. Also, what happens in the presence of the efflux-pump inhibitor CCCP?

11. In Fig. 4b the comparison between non-growers derived either from WT *L. pneumophila* or from a strain mutated in a type-IV secretion system suggests that the avirulent mutant strain dies more inside amoebae. How does this result apply to the subpopulation of growers? Also, what is the quantification of P4C-GFP enrichment for *D. discoideum* infected with growers? (Fig. 4d). To what extent the growers are less infectious than non-growers?

12. In Fig. 4e-g, by using two new fluorescent reporter strains, which are incompatible with the Timerbac system, the authors claim that intracellular non-growers upregulate the major flagellum subunit and an Icm/Dot-translocated effector. Here it is not clear how the authors could unambiguously identify the intracellular non-growers, only on the basis of static snapshots and in the absence of Timerbac, which they previously used as a proxy for growth. The Timerbac system, for instance, is compatible with blue and far-red fluorescent markers, which could be fused to PflaA and PsidC instead of GFP. The same remark applies to Fig. 5c-e and Lines 279-282.

13. Is non-growing *L. pneumophila* derived from activated macrophages capable of recovering growth in vitro, similar to what was shown in amoebae? Are these non-growers viable and not merely dead (similar to comment 6)? The same remark applies to Fig. 6.

14. An unequivocal definition of the status of intracellular persisters (e.g., line 290) is needed.

15. What is the growth rate of the different mutants in vitro during the growth curve compared to the WT? Minor, Fig. 6f is mislabeled in the text.

16. Is there a reason for testing only ofloxacin in Fig. 6d? Why the drug susceptibility assay was not carried at the subpopulation level?

Reviewer #1

Remarks to the Author:

“Quorum sensing elicits a subpopulation of vacuolar virulent *Legionella* persisters”, by Personnic and colleagues describes an experimental investigation of growth heterogeneity in *Legionella pneumophila* populations that had infected amoeba and murine macrophages. Using the Timer protein, the authors observed two distinct subpopulations in terms of fluorescence, which they assessed were growing and non-growing cells. The intracellular non-growing cells remained culturable and they exhibited higher tolerance to 3x MIC concentrations of antibiotics over short treatment periods (1 hr). The authors then measured the proteomes of growers and non-growers and discussed several differences. The non-growers happened to be quite virulent upon amoeba re-infection experiments, which was quite interesting. The authors found that the abundance of cells that expressed fluorescent virulence markers approximated the size of the population that was non-growing inside amoeba, and they observed similar phenomena in murine macrophages. Genetically, the authors implicated the type 4 secretion system, *lqsA* (an autoinducer synthase), both autoinducer sensory histidine kinases (*lqsS* and *lqsT*), and the transcription factor *lvbR*. Overall, an important topic with some interesting single cell data, and my concerns are detailed below.

Response: We would like to thank the reviewer for the overall positive evaluation of our study and the constructive suggestions, which strengthen the data about *L. pneumophila* persistence, the virulence of the intracellular non-growers and the role of the Lqs system.

Major comments:

1. To claim that the non-growing cells are persisters, higher antibiotic concentrations need to be used and biphasic kill curves need to be presented. Persister measurements are routinely conducted at very high antibiotic concentrations (sometimes up to 100x MIC) to prevent measurements from reflecting the abundance of spontaneously resistant mutants. Biphasic kill curves are also a requirement, which means survival needs to be measured as a function of time (not just at one time point) and two clear regimes must be observed when a log-linear plot of survival vs time is produced. The authors might find the following review helpful (PMID: 27080241). The authors should also assess whether survivors from those antibiotic treatments exhibit the same survival kinetics during a follow-up experiment (which has typically been done to show the phenotypic rather than genetic nature of persistence).

Response: The mean MIC for ofloxacin is 0.07 µg/mL (Stout JE et al. 1998; PMID 9488830 for erythromycin 0.5 µg/mL (Gómez-Lus et al. 2001; PMID 11463526), and for gentamicin 0.4 µg/mL (Havlichek et al. 1987; PMID 3435101), respectively. In our *ex vivo* antibiotic tolerance assay using sorted subpopulations (**Fig. 2e**), we thus used antibiotic concentration $\geq 10\times$ MIC. The text (**l. 186, l. 714**) was corrected accordingly.

To answer the reviewer's question, we performed antibiotic killing kinetics on the lysates of amoebae infected for 24 h with *L. pneumophila*/Timer using high antibiotic concentration ($> 100\times$ MIC). Biphasic kill curves were obtained confirming the presence of persisters in the infected cell lysates. Survivors from the antibiotic treatments were identified as non-growers according to the Timer color ratios. Importantly, the survivors exhibited similar survival

kinetics during follow-up experiments. The phenotypic rather than the genetic nature of the persistence was thus demonstrated.

Similar experiments were performed with infected macrophages and led to the intriguing discovery that *L. pneumophila* persistence was enhanced upon IFN- γ treatment.

These new results are shown in the new figures **Fig. 2f**, **Supplementary Fig. 7** and **Supplementary Fig. 14** and outlined in the text (**l. 193-202**, **l. 342-347**, **l. 703-709**). We also inserted the reference cited by the reviewer (PMID 27080241).

2. Line 44: Persisters are not resistant, they are tolerant to antibiotics.

Response: Thank you for pointing out this mistake, which we corrected (**l. 45**).

3. Line 94: Did they authors mean “-0.3” for growth arrest. 0.3 appears to be within the growing population.

Response: Thank you for pointing out this mistake, which we corrected (**l. 106**).

4. The y-axis in 2b should go down to 0%, starting at 30% is a bit misleading.

Response: We changed the y-axis in the **Fig. 2b**, as requested by the reviewer

5. Fig 2C: how can the CFU per event be higher than 100%? Somewhat lower than 100% is reasonable since some events might be caused by noise for instance, but why would it be over 100%? More CFUs than events that were sorted?

Response: The data presentation in Fig. 2c was indeed confusing. In the new **Fig. 2c**, we indicated the plating efficiency after sorting ($[\text{CFU} \times 100] / [\text{number of sorted fluorescent particles}]$) without normalization. The text was edited (**l. 160**, **l. 802**).

6. Line 178, I believe the authors mean “nitrosative”

Response: Corrected as suggested (**l. 230**).

7. Line 179-189: Stating that those over-abundant proteins are markers of bacterial persistence is not accurate. Unless the cited research was done in *L. pneumophila*, which it was not, there is no evidence that efflux pumps, cyclic AMP, ATP, or other proteins discussed are markers of persistence in *L. pneumophila*.

Response: We agree with the reviewer’s comment, down-toned the statement and moved the paragraph to the Discussion section (**l. 420-430**). Further research will address the unknown mechanisms driving *L. pneumophila* persistence in phagocytes using the proteomics data.

8. Line 162-200: None of the proteins mentioned in this section about comparative proteomics were actually investigated further. As far as I can tell, all of the statements in this section are speculative and not confirmed or explored with additional assays. It seems that this section would be better suited for the discussion, since they are discussing potential reasons why they would observe that data and not going further.

The section after this one with T4SS hits from the proteomics were followed up with additional assays, which justifies its placement in the results section.

Response: As suggested by the reviewer, we transferred the most speculative comments to the Discussion section (see point #7). We kept in the result section only proteins known to be a hallmark for growth or growth arrest as well as for metabolic routes and stress induction.

9. In Figure 4c, I can't see the wild-type growing cells being in those fluorescently labeled vacuoles. Perhaps because they are also labeled with GFP? Could a different fluorescent protein (not green or red, which Timer uses) be used for the vacuoles?

Response: In order to more easily distinguish the Timer-producing, replicating wild-type cells bacteria from the GFP-labelled LCV membrane, we highlighted the latter with white arrows in **Fig. 4c**. Thus, despite similar spectral properties, the morphological differences between the rod-shaped bacteria and the LCV membrane bilayer can be discriminated. The most important information conveyed in **Fig. 4c** is that the non-growing subpopulation of wild-type *L. pneumophila* but not $\Delta icmT$ mutant bacteria reside in a GFP-calnexin- or P4C-GFP-positive compartment, which can be readily observed with the fluorescent markers used. Due to the superior spectral properties, we routinely use in the lab GFP-calnexin or P4C-GFP to label the replication-permissive LCV, e.g., Weber et al., 2014, *mBio* 5:e00839-13; Weber et al., 2018, *mBio* 9: e02420-18. Moreover, to strengthen our arguments, we now also clearly show that among wild-type *L. pneumophila* residing in a P4C-GFP-positive compartment, only non-growing bacteria induce hallmark genes of motility and virulence (**new Fig. 5f**).

10. When looking at the *PsidC* and *PflaA-gfp* reporters, why didn't the authors change the fluorescent protein color so that they could assign cells as growing or non-growing with Timer, and then assess their expression of virulence factors in the same cells? Providing data where the abundance of cells that express those factors is close to the abundance of cells that are non-growing is not direct evidence of the connection between those two phenotypic outputs and growth status.

Response: We would like to thank the reviewer (and also reviewer #3, point 12) for this suggestion. As requested, we cloned the dual reporter construct constitutively producing Timer and the blue fluorescent protein mCerulean under control of promoters of hallmark genes of motility (*P_{flaA}*) or virulence (*P_{sidC}*) (*P_{tac-timer}* - *P_{flaA/sidC}*-*mCerulean*) (**Supplementary Fig. 10**). We used the construct to assess in *A. castellanii* (**new Fig. 5a-b**) as well as in macrophages (**Supplementary Fig. 15**) the growth state of individual bacteria, along with the expression of *P_{flaA}* or *P_{sidC}*. Using these reporters, we unambiguously showed that both the *flaA* and *sidC* promoters were specifically induced in the intracellular non-growing subpopulation, while *timer* was constitutively expressed. The corresponding results are outlined on **l. 289-305, l. 348-352, l. 545-550, l. 857-862.** respectively. The analysis made by confocal microscopy could not be quantified by flow cytometry, due to the lack of a flow cytometer with the right setup to detect blue fluorescence. Moreover, the use of a far-red fluorescent protein (e.g., mPlum) was not possible due to significant spectral overlapping with Timer.

Moreover, infection of *D. discoideum* producing the Golgi/LCV PtdIns(4)*P* probe P4C-GFP, for 24 h, confirmed that intracellular non-growers produced mCerulean and resided in LCVs

(new Fig. 5f). The corresponding results are outlined in the revised manuscript (l. 320-322, l. 870-875).

11. Figure 6f: can the authors provide an explanation as to why deletion of both sensory kinases produced the opposite effect of the single deletions?

Response: LAI-1-dependent signalling is complex and proceeds through the two homologous sensor kinases LqsS and LqsT, which are encoded by the *lqsS* gene located in the *lqs* cluster and the orphan *lqsT* gene, respectively (Kessler et al., 2013, *Environ Microbiol* 15:646-662). The virulence and other phenotypes of the $\Delta lqsS$ - $\Delta lqsT$ double mutant strain are partially complemented by providing either *lqsS* or *lqsT* on a plasmid. This feature is also seen for reversion of the phenotype of the $\Delta lqsS$ - $\Delta lqsT$ mutant regarding the increased ratio of growers vs. non-growers (new Fig. 7f). Moreover, the *lqsS* and *lqsT* genes are differentially regulated in stationary growth phase, and transcriptome studies indicated that 90% of the genes, which are downregulated in absence of *lqsT*, are upregulated in absence of *lqsS* (Kessler et al., 2013, *Environ Microbiol* 15:646-662). The reciprocal and complex gene regulation pattern implicating signaling through LqsS and/or LqsT might also account for the fact that upon infection of *A. castellanii* with either the $\Delta lqsS$ or the $\Delta lqsT$ mutant strain the percentage of non-growers was lower compared to wild-type *L. pneumophila*, while upon infection with the $\Delta lqsS$ - $\Delta lqsT$ double mutant strain the percentage was higher (new Fig. 7f and Supplementary Fig. 17). These reflections are now outlined in the Discussion section (l.461-473).

12. Complementation of the *lqsA* deletion mutant was well received, but complementation of the other mutants with phenotypes would be fitting (*icmT*, *lqsS*, *lqsT*, *lvbR*).

Response: As requested by the reviewer, we complemented the strains $\Delta icmT$, $\Delta lqsS$, $\Delta lqsT$, $\Delta lvbR$ as well as $\Delta lqsS$ - $\Delta lqsT$. In order to complement the mutation while keeping the timer expression, we co-expressed on a same plasmid *timer* under control of the *P_{tac}* promoter rendered constitutive and the gene of interest under the control of its native promoter. The phenotypes of the mutant strains are shown along with the complementation in the new Fig. 7f and outlined in the text (l.39-396; l.558-561; l. 909-913). Complementation of the *icmT* mutant strain is shown in the new Fig. 4b, Supplementary Fig. 3, Supplementary Fig. 12 and outlined in the text (l. 130-131; l. 261-266; l. 329-330; l. 551-561; l. 841-845).

Reviewer #2

Remarks to the Author:

The manuscript by Personnic et al. "Quorum sensing elicits a subpopulation of vacuolar virulent *Legionella* persisters" describes how the non-replicating virulent persisters of *Legionella* in phagocytes are controlled by the Lqs system and what are the differences between the replicating and non-replication populations in the phagocytes and amoebae. I was asked to comment specifically on the proteomics part of the study.

It is difficult to be sure whether the study is technically sound because the data is shown in a way that precludes such insight. There are several modifications at least to the data presentation that have to be made before the manuscript can be reconsidered.

Response: We would like to thank the reviewer for the evaluation of our study and the constructive suggestions to improve the proteome analysis documented in the manuscript.

Major comments:

1. The raw data should be deposited in a publicly available database (such as ProteomeXchange) and the dataset access should be clearly described.

Response: The mass spectrometry proteomics data have been deposited to the ProteomeXchange Consortium via the PRIDE (Perez-Riverol et al. 2019. *Nucleic Acids Res* 47(D1):D442-D450, (PMID: 30395289) partner repository with the dataset identifier PXD015106 and 10.6019/PXD015106. Outlined **I. 621-623**.

Reviewer account details: Username: reviewer60623@ebi.ac.uk / **Password:** n1AXFNrh

2. It was difficult to see if there were any replicates, even though one could assume there were because the statistical analysis was performed. The only information about the three replicates was in the legend for the Supplemental Figure 2. It should be clearly stated in the Methods section.

Response: The FACS Sorting to generate the proteome of non-growers vs growers and slow-growers vs fast-growers was repeated 4 times. The four replicates were used individually to generate the proteome dataset slow-growers vs fast-growers. To enhance protein detection for the comparison of non-growers vs growers, we combined the four samples (sort 1 to 4) into two (sort 1+2 and sort 3+4). Indeed, sorting of intracellular bacterial subpopulations for subsequent proteomics analysis is a technically challenging approach, as the sorting time and the yield (a few millions for the purest sorting) is limiting. In similar studies, researchers have pooled all their replicates in one final sample to be processed for proteomics in order to increase protein detection (Claudi et al., 2014, *Cell* / PMID 25126781; Becker et al., 2006, *Nature* / PMID 16541065). We corrected the figure legends (Fig. 3 and Supplementary Fig. 8) and the Method section as requested (I. 566-567).

3. Were all the proteins listed in the tables identified and quantified in all three replicates? The tables should list the numbers for each replicate separately and then the averages and p-values.

Response: Yes. We modified **Table 1** and **Table 2** in order to separate the replicates and show the averages and p-values. Moreover, we modified **Fig. 3c** and **Fig. 3d** to accommodate the standard deviations.

4. Ultimately, the results of the proteomic analysis do not seem to show anything surprising or novel, but mainly confirm the known features of different subpopulations, or at least the data is presented that way.

Response: Our study represents the first report of distinct – growing and non-growing – intracellular *L. pneumophila* subpopulations in evolutionarily distant phagocytes (amoebae, macrophages). The finding that the non-growing subpopulation comprises viable, antibiotic-tolerant persisters, which are highly infectious is novel and unexpected, and has broad implications for the ecology, evolution and pathogenesis of *Legionella* species. The proteomics of different FACS-sorted intracellular *L. pneumophila* subpopulations is also novel in the field of *Legionella* research and technically challenging. The comparative proteomics analysis validates the presence of distinct subpopulations and identifies subpopulation-specific factors, which in future studies will be characterized in detail.

Reviewer #3

Remarks to the Author:

Legionellosis occurs worldwide and outbreaks are mainly caused by waterborne *Legionella pneumophila*, which is fatal if not treated promptly, and is the focus of this manuscript. *L. pneumophila* is ubiquitous in the environment, and can multiply both within aquatic protozoa and in macrophages. Moreover, the growth and viability of *L. pneumophila* can vary under environmental fluctuations, such as different temperatures.

The authors use a Timer fluorescent reporter system, constitutively expressed from an episomal plasmid, to monitor the growth state of *L. pneumophila* during infection of amoebae and of naïve and activated murine macrophages. The authors infer that *L. pneumophila*-derived either from amoebae or from activated macrophages produce about 50% of non-growing cells. The authors also tested whether the non-growing bacteria were viable, metabolically active, antibiotic susceptible, and virulent, and they made a comparative analysis of the proteome of both growing and non-growing subpopulations extracted from amoebae. They claim that phenotypic heterogeneity of growth among intracellular bacteria is reversible *in vitro*. They also conclude that non-growing bacteria tolerate better antibiotic treatment, express persistence-related proteins and are more virulent. At the end of the manuscript the authors, relying partly on their proteomic data and partly on the literature, create some *L. pneumophila* mutant strains, striving to understand the genetic origin of the intracellular formation of persistent variants, which remains open to speculation.

In conclusion, the topic addressed in this manuscript is scientifically relevant and ambitious. Nonetheless, the fluorescent reporter system used, on which almost all the experiments were based, was not sufficiently characterized. This makes it difficult to draw clear conclusions in a systematic manner, also because of the lack of some adequate controls. This raises major concerns, which are detailed below, that need to be addressed in order to support the main conclusions of the manuscript.

Response: We would like to thank the reviewer for the overall positive evaluation of our study and the constructive suggestions to improve the scientific and technical robustness of our work.

Major comments:

1. The Timerbac system was well characterized in *Salmonella* (Claudi et al., 2014, *Cell* 158: 722-733) and used as a growth-rate reporter. Although the Timerbac system is very attractive, several factors can influence its expression, including oxygen. Timerbac should be used cautiously in a new species, notably when the bacterium is versatile as to environmental changes, such as temperature. The authors should provide a careful characterization of this fluorescent reporter system in their model organism. In particular, the correlations between the growth rate and green and red fluorescence over the whole growth curve in vitro (OD and CFUs) and at different temperatures used to grow amoebae and macrophages (25°C and 37°C) are lacking. For instance, Fig. 1a shows the relationship between the Timer Log10 color ratio and two growth phases, without providing information regarding the actual division rate.

Response: We would like to thank the reviewer for these insightful comments. As suggested, in the revised manuscript, we now correlate the Timer color ratios along the growth curve at 37°C in AYE broth and removed any reference to the bacterial growth rate (**new Fig. 1a**, outlined in the text **l. 85-91, l. 768-769**). For the exact correlation between Timer color ratios and division rates at 37°C and 25°C, **please see our response to comment #2**.

Oxygenation is indeed a parameter influencing the Timer maturation kinetics. However, *L. pneumophila* is a strictly aerobic bacterium, and thus, all experiments were performed under normal atmosphere (+/- CO₂ for the macrophages). The temperature is another parameter to take into account. Yet, within our range of working temperatures, DsRed-derivative maturation kinetics are expected to be only moderately affected (Verkhusha, V. V. *et al.*, 2004, PMID 15217617).

2. Since the G/R ratio was not clearly associated to specific growth rates, how were the non-growers exactly defined (Fig. 1d, 4, 5, 6, Supplementary Figures 1a, 3, etc.)? For instance, in lines 596-97 and 409 and in SupplInfo line 28, the authors claim that non-growing bacteria appear red, whereas the bacteria look mostly yellowish/orange as they still express green fluorescence. Could they be slow growers and what is the intracellular growth rate of cells that turned green? The definition of NG, Gs and Gf should be carefully determined in vitro first and then applied to the infection models.

Response: We agree that the division rate of the non-growers, the slow-growers and the fast-growers should be defined. To address the reviewer's concern (also regarding comment #1), we immobilized *L. pneumophila*/Timer in AYE/agarose and followed the bacterial division over time by confocal microscopy at 25°C and 37°C. The Timer color ratio was measured for each individual cells, and was correlated precisely to the growth rate (number of cell divisions that occurred per unit of time). Thus, we found that the correlation between the Timer color ratios and the growth rate was rather insensitive to temperature changes. The values obtained *in vitro* (in broth) were also validated *in vivo* (in the amoebae infection model). We empirically equate the correlation between the growth rate and the Timer color ratio as follows: $[\mu = \frac{0.1 + \text{Log}_{10}R}{2.6}]$, where μ is the growth rate and R the Timer color ratio 500nm/600nm. This dataset is presented in the **new Supplementary Fig. 1b and c, Fig. 1b and c, Supplementary Fig. 2c** and outlined in the text **l. 91-97; l. 102-106; l. 113-117, l. 636-642; l. 776-777**.

More generally spoken, the visualization of the Timer color ratios is inextricably linked to the acquisition devices. In this study, we used confocal microscope, flow-cytometer, sorter, and

image-flow cytometer. For the flow-cytometry-based analysis, the non-growers were determined according to and based on the spectral properties of the avirulent *L. pneumophila* $\Delta icmT$ mutant strain, which lacks a functional Icm/Dot T4SS, cannot replicate intracellularly, and thus, by definition is non-growing. A detailed subset gating strategy is shown in the response to comment #7 (**please see below**).

In very sensitive approaches such as flow-cytometry, the non-growers have a distinct, well defined and non-biased low green/red fluorescence ratio, well separated from the growing individuals (**see Fig 1c**). The same applies to confocal microscopy, when it comes to exact fluorescence measurements. However, the human eye may have difficulties to discriminate between those ratios on micrographs with overlaying fluorescence signal and bright field. We consider the color ratio range for non-growers from red to yellowish/orange. Indeed the non-growers harbor GFP, as they are metabolically active and keep producing Timer as well as other subpopulation-specific proteins (see the FRAP experiments and the proteome). Importantly, intracellular *L. pneumophila* replicate within a LCV. Thus non-growers appear as isolated individuals, whereas proliferating bacteria will form membrane-bound clusters.

3. Moreover, since Timerbac was expressed from an episomal plasmid, it is important to assess to what extent the variable number of copies of the plasmid contributes to the observed heterogeneity.

Response: The plasmid pMMB207C used throughout the study is a medium copy expression vector suitable for *L. pneumophila*. Plasmid copy numbers can indeed vary at a single cell-level and due to gene-dose effects may interfere with the overall number of Timer molecules produced per cell (regardless of their maturation state). However, the kinetics of Timer maturation is the same in all cells, and the plasmid copy does not impact the Timer color ratio. In **Fig. 1a**, the peak sharpness indicates that the fluorescence ratios are robust against cell-to-cell variations in protein content and bacterial cell size. In the revised version of the manuscript, we now comment on this aspect (**l. 91-93**).

4. It is also unclear how the bacteria behaved during the first 5 hours of infection (Fig 1b and Movie) and whether they were already non-growing prior to infection, or if they transitioned to a non-growing state upon infection.

Response: To grow *L. pneumophila* as an inoculum for infection experiments, we rigorously use a canonical protocol in the lab: 3 ml of AYE are inoculated at an OD₆₀₀ of 0.1 and grown fully aerated at 37°C for 21 h. Under these conditions, the bacterial culture robustly and reproducibly comprises stationary phase bacteria, which are virulent and motile (as observable by microscopy), i.e. these bacteria are in the “transmissive” state. *L. pneumophila* employs a biphasic life style and within the first hours of infection switches from the transmissive to the replicative state. Hence, during the first 5 h of infection in the replication-permissive LCV, this switch takes place.

5. A reference for the $\Delta icmT$ strain is missing, as well as a comparative growth curve between *L. pneumophila* WT, $\Delta icmT$ and the strain overexpressing Timerbac.

Response: The reference for the $\Delta icmT$ strain has been added (**l. 128, Table 3**). The requested growth curves were performed and added as **new Supplementary Fig. 1a** (text: **l. 84-85**). The

calculated exponential growth rates are indicated and similar for all strains. The $\Delta icmT$ has a shorter lag phase, known for long in the *Legionella* field.

6. To determine the viability of non-growing *L. pneumophila* the authors assessed the membrane potential by staining bacteria with a far-red dye, which is mainly used to label mitochondria. Indeed the BacLight Bacterial Membrane Potential Kit would be a more specific tool, even though not compatible with Timerbac. Moreover, they claim that the presence of green fluorescence in non-growing cells, which are often defined as red, is an indicator of active metabolism. Being the Timerbac reporter expressed from a strong constitutive promoter, the green fluorescence, even in the absence of growth, could depend on the accumulation of older protein, which is not matured into its red form, due to the absence of metabolic activity. A FRAP experiment should help solving this issue. In addition, non-growing cells might be dead even though they continue to glow with fluorescence, because old protein expressed from a strong promoter can endure in the body of dead cells. In sum, it is crucial to perform counter-staining to spot live versus dead cells, and this must also be associated with the sorting experiments throughout the manuscript.

Response: As suggested by the reviewer, we performed FRAP experiments and demonstrated that non-growers indeed synthesize the Timer protein *de novo* (**new Fig. 1f**, outlined in **I. 145-148, I. 644-651** and **I. 784-792**). Moreover, we also demonstrate that the non-growing subpopulation is viable by performing a live/dead propidium iodide staining with lysates of amoeba infected with GFP-producing *L. pneumophila*. We found that less than 5% of the bacteria were dead (**new Supplementary Fig. 4**, outlined in the text **I. 139-141, I. 667-674**). Under our experimental conditions, bacterial death is thus neglectable. Finally, the non-growing subpopulation produces a specific proteome (**Fig. 3**), in agreement with an active metabolism and alternative physiology compared to the growing subpopulation.

7. Although Fig. 2b shows a good sorting efficiency, the gating strategy with regard to both a non-fluorescent control strain and the sorted subsets must be specified, and the same number of events per subset should be shown in Fig. 2a and Supplementary Figure 2.

Response: As requested, the gating strategy to identify the bacterial subpopulations for subsequent FACS-sorting is now detailed in the **new Supplementary Fig. 2b and c** (text **I. 108-117**). We also included the avirulent $\Delta icmT$ mutant strain. The **Supplementary Fig 8a** (**former supplementary Fig. 2a**) was adapted as requested.

8. In the sorted subset of growers (Fig. 2a) there is a fraction of cells that has a fluorescence ratio comparable to the non-growers. How did the authors make sure that the growth resumption found in non-growers (Fig. 2c,d) was not due to a contamination from growers and vice versa?

Response: The sorted non-growers are nearly pure, and the contamination of the non-growers is below 5%. Thus, ~ 95% of the sorted population belongs to a distinct subpopulation and should not affect subsequent CFU quantification and harvesting. To experimentally demonstrate the growth resumption capacity of the non-growers, we used sorted non-growers

to infect fresh amoeba and visualized the intracellular growth resumption over time by confocal microscopy (new **Supplementary Fig. 5**, new **Supplementary movie 2**). These result are outlined in the text (**l. 163-169**).

9. In lines 137-140 the authors claim that the formation of growing and non-growing subpopulations is a reversible phenomenon, and that the sorted subsets developed similar percentages of growers and non-growers. Next, in lines 145, 146, 628 and 632, the authors state that they tested the sensitivity of “intracellular” *L. pneumophila* to antibiotics, and claim that “intracellular” non-growers are non-replicating drug persisters. However, the persistence assay was not carried out on “intracellular” bacteria but on subpopulations extracted from the host, which apparently recover a similar phenotype in vitro. To test which of the two subpopulations survives drug treatments best, the authors should directly treat the intracellular bacteria with antibiotics and extract and sort them only after treatment. Therefore lines 152, 153, 159, 160 and 179 are not corroborated by the experiments.

Response: As requested by the reviewer, we exposed *A. castellanii* infected with *L. pneumophila* 24 h p.i. to high concentrations of ofloxacin (300 µg/mL, 1 h) prior to FACS-sorting and CFU quantification of bacterial subpopulations. Thus, we confirmed that non-growers better survive antibiotics treatment *in vivo* (new **Supplementary Fig. 6a and b**). We also specify that the experiment **Fig 2e** was performed *ex vivo*. These changes are now also outlined in the text (**l. 189-192, l. 710-721 and l. 811**).

10. The authors propose that increased antibiotic efflux in non-growers is likely to cause antibiotic persistence. However, it seems more rational that cells that do not grow are less efficient at internalizing compounds, as shown in Fig. 2f, rather than at extruding them more. Also, what happens in the presence of the efflux-pump inhibitor CCCP?

Response: At this point, we cannot discriminate between increased antibiotic efflux and decreased antibiotic influx. We account for this fact in the text (**l. 205-212**): “In agreement with a potential contribution to antibiotic tolerance, we measured significantly lower drug acquisition in FACS-sorted non-growing compared to growing *L. pneumophila* in lysates of infected *A. castellanii* (**Supplementary Figure 7b, former Fig. 2f**). ... intracellular *L. pneumophila* non-growers are defined as non-replicating persisters with increased drug efflux capacity (or reduced drug accumulation)”.

The protonophore CCCP is not a specific efflux pump inhibitor, but will rather dissipate the membrane potential of prokaryotes as well as eukaryotic organelles and cells. Thus, experiments with CCCP will be inherently unspecific and inconclusive.

11. In Fig. 4b the comparison between non-growers derived either from WT *L. pneumophila* or from a strain mutated in a type-IV secretion system suggests that the avirulent mutant strain dies more inside amoebae. How does this result apply to the subpopulation of growers? Also, what is the quantification of P4C-GFP enrichment for *D. discoideum* infected with growers? (Fig. 4d). To what extent the growers are less infectious than non-growers?

Response: As also requested by reviewer #1, we modified **Fig. 4b** in order to complement the $\Delta icmT$ phenotype. The results are also outlined in the text (**l. 261-266** and **l. 841-845**). The lack of a functional Icm/Dot T4SS system prevents the $\Delta icmT$ mutant strain from forming a replicative subpopulation, and thus, $\Delta icmT$ produces only non-growers intracellularly.

For **Fig. 4d**, we carried out a new series of experiments and quantified the P4C-GFP enrichment with the growers. The result is outlined also in the text (**l. 277-279**).

As shown in **Fig. 4a**, growers cannot resume growth upon re-infection (when not passaged on plate and re-grown for subsequent infection). Moreover, the new **Fig. 5f** (**l. 320-322; l. 870-875**) indicates that only the non-growers expressed the virulence gene *sidC* and that they built a protective *Legionella* containing vacuole.

12. In Fig. 4e-g, by using two new fluorescent reporter strains, which are incompatible with the Timerbac system, the authors claim that intracellular non-growers upregulate the major flagellum subunit and an Icm/Dot-translocated effector. Here it is not clear how the authors could unambiguously identify the intracellular non-growers, only on the basis of static snapshots and in the absence of Timerbac, which they previously used as a proxy for growth. The Timerbac system, for instance, is compatible with blue and far-red fluorescent markers, which could be fused to PflaA and PsidC instead of GFP. The same remark applies to Fig. 5c-e and Lines 279-282.

Response: As for reviewer #1 (point 10), we would like to thank the reviewer for this suggestion. As requested, we cloned the dual reporter construct constitutively producing Timer and the blue fluorescent protein mCerulean under control of promoters of hallmark genes of motility (P_{flaA}) or virulence (P_{sidC}) ($P_{tac-timer} - P_{flaA/sidC} - mCerulean$) (**Supplementary Fig. 10**). We used the construct to assess in *A. castellanii* (**new Fig. 5a-b**) as well as in macrophages (**Supplementary Fig. 15**) the growth state of individual bacteria, along with the expression of P_{flaA} or P_{sidC} . Using these reporters, we unambiguously showed that both the *flaA* and *sidC* promoters were specifically induced in the intracellular non-growing subpopulation, while *timer* was constitutively expressed. The corresponding results are outlined on **l. 289-305, l. 348-352, l. 545-550, l. 857-862.**, respectively. The analysis made by confocal microscopy could not be quantified by flow cytometry, due to the lack of a flow cytometer with the right setup to detect blue fluorescence. Moreover, the use of a far-red fluorescent protein (e.g., mPlum) was not possible due to significant spectral overlapping with Timer.

Moreover, infection of *D. discoideum* producing the Golgi/LCV PtdIns(4)P probe P4C-GFP, for 24 h, confirmed that intracellular non-growers produced mCerulean and resided in LCVs (**new Fig. 5f**). The corresponding results are outlined in the revised manuscript (**l. 320-322, l. 870-875**).

13. Is non-growing *L. pneumophila* derived from activated macrophages capable of recovering growth in vitro, similar to what was shown in amoebae? Are these non-growers viable and not merely dead (similar to comment 6)? The same remark applies to Fig. 6.

Response: To experimentally demonstrate the growth resumption capacity of the non-growers deriving from IFN- γ treated macrophages, we used sorted non-growers to infect fresh *A. castellanii* amoeba and visualized the intracellular growth resumption over time by confocal

microscopy (**new Supplementary Fig. 13**). The result are outlined also in the text (l. **338-341**). Finally, intracellular $\Delta lqsA(P_{sidC-gfp})$ non-growers specifically produce GFP, in agreement with active metabolism and bacterial viability (**Supplementary Fig. 16c, l. 382-383**).

14. An unequivocal definition of the status of intracellular persisters (e.g., line 290) is needed.

Response: As suggested, we now define intracellular persisters (**l. 193-195**).

15. What is the growth rate of the different mutants in vitro during the growth curve compared to the WT? Minor, Fig. 6f is mislabeled in the text.

Response: The growth curves were performed in AYE broth and added as **new Supplementary Fig. 18b** and in the text (**l. 397-398**). The $\Delta lqsR$ mutant strain has a shorter lag phase, previously described in Tiaden *et al.*, 2007, PMID 17614967. The calculated exponential growth rate is indicated and similar for all strains. During the infection, the median Timer color ratio (R) for the growing subpopulation was determined by flow cytometry to estimate the growth rate (μ) using the empirical formula [$\mu = \frac{0.1 + \text{Log}_{10}R}{2.6}$] (**new Supplementary Fig. 18a and new supplementary Fig. 1a, b**). The growth rate is the same for all tested strains. The result is outlined in the text (**l. 396-397**). We corrected the figure labeling as requested.

16. Is there a reason for testing only ofloxacin in Fig. 6d? Why the drug susceptibility assay was not carried out at the subpopulation level?

Response: We tested only ofloxacin, as it previously showed the highest bactericidal activity in our persister assays (**Fig. 2e**). The drug susceptibility assay was performed at the population level to detect differences in the antibiotic susceptibility of the bacterial strain (WT or $\Delta lqsA$) producing less persisters.

REVIEWERS' COMMENTS:

Reviewer #2 (Remarks to the Author):

The authors have adequately addressed my concerns.

Reviewer #3 (Remarks to the Author):

The authors have addressed the concerns related to the first submission.

Reviewer #4 (Remarks to the Author):

In "Quorum sensing elicits a subpopulation of vacuolar virulent *Legionella* persisters," Personnic and colleagues report growth heterogeneity in *Legionella pneumophila* populations in amoeba and macrophages. The authors show that non-replicating cells in the phagocytes were culturable, retained metabolic activity, and were more tolerant to antibiotics. They also show that *L. pneumophila* can form persisters in amoeba and macrophages. Using comparative proteomics, the authors found that growers and non-growers exhibit distinct proteomes. They also found that non-growers are more virulent compared with growing cells. Using genetic mutants, they link the type IV secretion system and the Lqs pathway to the formation of these virulent non-growers.

The authors addressed the main concerns raised by Reviewer 1 from the previous revision. In particular, they showed in *ex vivo* assays that non-growers were tolerant to higher concentrations (10X MIC) of antibiotics, and they generated biphasic kill curves using antibiotics administered at 100X MIC to show that *L. pneumophila* from amoeba lysates formed persisters. They demonstrated that the survivors exhibited similar killing kinetics as the original population, confirming the phenotypic nature of tolerance. They also engineered dual-color reporters to monitor the growth and expression of *flaA*/*sidC* simultaneously. Furthermore, they genetically complemented mutants of genes in the LAI-1 pathway to establish their role in the formation of intracellular non-growers.

Upon reviewing their manuscript and the response to reviewers, I have a few minor comments, which are indicated below:

1. The results presented in Supplemental Fig. 7A may be misleading/ confounding. Using flow cytometry, the authors show that cells remained after 50 h of treatment with erythromycin or ofloxacin administered at 100X MIC were non-growers. Unlike resistant mutants, persisters and non-persisters are not expected to grow in the presence of antibiotics at concentrations above the MIC. Further, recent studies have suggested that cell death may not necessarily occur during antibiotic treatment; instead, it occurs after antibiotic removal (e.g., Barrett et al., 2019, Nature Comm. 10: 1177; Hong et al., 2019, PNAS 116: 10064-10071). Without plating the non-growers to ensure that they can form colonies, it is difficult to distinguish them from persisters/ survivors from cells that are destined to die.

2. When describing the methods for biphasic kill curves in the Materials and Methods section, the authors should indicate the number of times the bacteria were washed, as well as the volume of supernatant removed and wash solution added back each time. This is to ensure that the antibiotic concentrations are well below the MIC following these washes.

3. Lines 420-430: Although the authors moved their comments on abundant proteins found in the non-growers (e.g., efflux pumps, cyclic AMP, and ATP synthase) to the discussion section and mentioned that they have toned down these statements, calling these “markers of persistence” without experimental support is still inaccurate. Perhaps they can just state that proteins that have been shown to be important in persisters originating from other bacteria are elevated in the proteome of *L. pneumophila* non-growers.

4. Line 61: “life style” should be changed to “lifestyle.”

Reviewer #2

Remarks to the Author:

The authors have adequately addressed my concerns.

Response: We would like to thank the reviewer for the overall positive evaluation of our study.

Reviewer #3

Remarks to the Author:

The authors have addressed the concerns related to the first submission.

Response: We would like to thank the reviewer for the overall positive evaluation of our study.

Reviewer #4

Remarks to the Author:

In “Quorum sensing elicits a subpopulation of vacuolar virulent *Legionella* persisters,” Personnic and colleagues report growth heterogeneity in *Legionella pneumophila* populations in amoeba and macrophages. The authors show that non-replicating cells in the phagocytes were culturable, retained metabolic activity, and were more tolerant to antibiotics. They also show that *L. pneumophila* can form persisters in amoeba and macrophages. Using comparative proteomics, the authors found that growers and non-growers exhibit distinct proteomes. They also found that non-growers are more virulent compared with growing cells. Using genetic mutants, they link the type IV secretion system and the Lqs pathway to the formation of these virulent non-growers.

The authors addressed the main concerns raised by Reviewer 1 from the previous revision. In particular, they showed in *ex vivo* assays that non-growers were tolerant to higher concentrations (10X MIC) of antibiotics, and they generated biphasic kill curves using antibiotics administered at 100X MIC to show that *L. pneumophila* from amoeba lysates formed persisters. They demonstrated that the survivors exhibited similar killing kinetics as the original population, confirming the phenotypic nature of tolerance. They also engineered dual-color reporters to monitor the growth and expression of *fla*/*sidC* simultaneously. Furthermore, they genetically complemented mutants of genes in the LAI-1 pathway to establish their role in the formation of intracellular non-growers. Upon reviewing their manuscript and the response to reviewers, I have a few minor comments, which are indicated below:

Response: We would like to thank the reviewer for reviewing our work and for the overall positive evaluation of our study.

1. The results presented in Supplemental Fig. 7A may be misleading/ confounding. Using flow cytometry, the authors show that cells remained after 50 h of treatment with erythromycin or ofloxacin administered at 100X MIC were non-growers. Unlike resistant mutants, persisters and non-persisters are not expected to grow in the presence of antibiotics at concentrations above the MIC. Further, recent studies have suggested that cell death may not necessarily occur during antibiotic treatment; instead, it occurs after antibiotic removal (e.g., Barrett et al., 2019, *Nature Comm.* 10:1177; Hong et al., 2019, *PNAS* 116:10064-10071). Without plating the non-growers to ensure that they can form

colonies, it is difficult to distinguish them from persisters/ survivors from cells that are destined to die.

Response: Persisters are often described as non-growing bacteria. However drug tolerance had also been detected in replicating mycobacterial species (Adam, K.N., Cell, 2011; Wakamoto, Y., Science, 2013). The Supplemental Fig.7a shows the lack of bacterial replication upon antibiotic exposure. The flow-cytometry analysis was carried out in parallel to the killing kinetics presented in Fig. 2f (which is based on CFU numbering). Altogether, we can thus conclude that the antibiotic survivors, that resumed growth on plates, originated from non-replicating bacteria. We corrected the text (**l. 200-202 and the supplemental figure legend 7a**) to avoid any misinterpretation or misunderstanding. We also thank the reviewer for the two recent and very nice studies she/he highlighted.

2. When describing the methods for biphasic kill curves in the Materials and Methods section, the authors should indicate the number of times the bacteria were washed, as well as the volume of supernatant removed and wash solution added back each time. This is to ensure that the antibiotic concentrations are well below the MIC following these washes.

Response: We modified the Methods section to integrate the reviewer concerns. (**l. 710-712**) “At given time points, bacteria were collected and washed 3 times. For each wash 1 mL PBS was used and 950 μ L of the supernatant discarded. The bacterial suspensions were subsequently serially diluted 1:10 and plated to quantify CFUs”. After the washes, the serial dilutions and the plating, we estimate that the final concentration in antibiotic is about 0.00001 ng/mL and thus neglectable.

3. Lines 420-430: Although the authors moved their comments on abundant proteins found in the non-growers (e.g., efflux pumps, cyclic AMP, and ATP synthase) to the discussion section and mentioned that they have toned down these statements, calling these “markers of persistence” without experimental support is still inaccurate. Perhaps they can just state that proteins that have been shown to be important in persisters originating from other bacteria are elevated in the proteome of *L. pneumophila* non-growers.

Response: We agree with the reviewer’s comment and modified our statement as suggested (**l. 421-423**).

4. Line 61: “life style” should be changed to “lifestyle.”

Response: Corrected as suggested (**l. 61**).